# DecFus: Decentralized Layer-wise Fusion with Dynamic Exploration and Exploitation

Li Yang [* 1]  Jialong Sun [* 2]  Chuhai Cai [2]  Xinyang Liu [3]  Yichen Li [4]  Bowen Peng [5]  Jialong Li [2]  Bo Liu [2]

## Abstract

Decentralized Federated Learning (DFL) enables collaborative model training across connected clients without a central server, mitigating communication bottlenecks and avoiding the single point failure in Centralized Federated Learning (CFL) . However, existing DFL methods mostly focus on parameter averaging with compromised update directions, which limits their potential due to insufficient exploration of the loss landscape, especially for complex models. We observe that layer exchanges among clients enhance exploration while introducing instability due to highly diverse update directions. To address these limitations, we propose Decentralized Layer-wise Fusion (DecFus), the first DFL framework that unifies layer-level exchange and averaging to balance exploration and exploitation. DecFus dynamically transitions the decentralized training process from exploration-dominant to exploitation-dominant phases, guided by the loss variance among connected neighbors. Furthermore, a layer-wise fusion strategy, informed by pairwise cosine similarity, categorizes all layers into two groups: an exchange group for exploration and an averaging group for exploitation. Additionally, we theoretically establish the convergence of DecFus without relying on the common assumption in existing literature that the aggregation matrix must be doubly stochastic. Extensive experiments verify that DecFus outperforms existing CFL and DFL schemes on different datasets under various settings.

## 1. Introduction

Centralized Federated Learning (CFL) relies on a central server to achieve collaborative model training across diverse clients without sharing local data (McMahan et al., 2017; Liu et al., 2023), but its reliance on a central server causes communication bottleneck and single-point failure (Liu et al., 2020; Li et al., 2021). Decentralized Federated Learning (DFL) (Liu et al., 2025b) mitigates these issues through direct client-to-client communication (Sun et al., 2024; Liu & Ding, 2021b), yet still suffers from degraded performance under non-IID data due to biased aggregation and slow information propagation among neighbors, which is further exacerbated by sparse communication topology (Sun et al., 2023; Shi et al., 2023). Although various methods have been proposed to address this issue through guided local training or adjusted aggregation weights (Jiang et al., 2022; Liu et al., 2024), they share a common limitation of the reliance on model averaging paradigm, fundamentally limiting their performance.

Conventional model averaging paradigm suffers from two major limitations: ignorance of local knowledge and the stuck-at-local-search problem(Hu et al., 2024). These issues stem from insufficient exploration during the model aggregation, leading to suboptimal generalization, particularly in large, complex neural network models which have highly non-convex loss landscape filled with numerous saddle points and local minima(Ly & Gong, 2025; Keskar et al., 2017). Empirical studies reveal that models with better generalization typically converge to wide, flat valleys, while sharp minima lead to inferior robustness (Kwon et al., 2021).

To address the exploration deficiency of model averaging, recent works introduce parameter perturbations and layer exchanges, outperforming naive averaging in navigating the loss landscape (Hu et al., 2024; Caldarola et al., 2022). However, these methods mainly focus on identifying promising directions or flatter regions in the landscape through a holistic view, while overlooking the intrinsic optimization dynamics of model itself, including the temporal evolution of training and the heterogeneous sensitivity of parameters across different layers. Such dynamics are known to be fundamental for effective learning and generalization, as evidenced by classical adaptive optimization methods (Duchi

[*]Equal contribution  [1]Department of Computer Science, Durham University, Durham, UK [2]Faculty of Computer Science and Artificial Intelligence, Shenzhen University of Advanced Technology, Shenzhen, China [3]School of Engineering Mathematics and Technology, University of Bristol, Bristol, UK [4]School of Computer Science and Technology, Huazhong University of Science and Technology, Wuhan, China [5]Department of Network Intelligence Research, Pengchen Laboratory, Shenzhen, China. Correspondence to: Bo Liu <liubo@suat-sz.edu.cn>.

*Proceedings of the $43^{rd}$ International Conference on Machine Learning*, Seoul, South Korea. PMLR 306, 2026. Copyright 2026 by the author(s).

et al., 2011; Kingma & Ba, 2014).

Motivated by the above insights, we propose a novel **Decentralized Layer-wise Fusion (DecFus)** framework that unifies layer averaging and exchange while explicitly considering the dynamic nature of the training process. This approach enhances DFL performance and overcomes the limitations of naive model averaging. Specifically, layer exchange serves as exploration for exploring larger region and escaping from sharp ravine, while layer averaging constitutes exploitation for steering the model toward more stable and potentially better solutions. DecFus dynamically balances these two behaviors during training progresses by adaptively orchestrating layer-wise exchange and averaging, achieving an effective exploration–exploitation trade-off at the layer level. This allows the DFL system to escape suboptimal minima and converge to superior solutions. We further extend DecFus to Efficient Decentralized Layer-wise Fusion (EDecFus), combining with a layer selection mechanism to balance communication efficiency and model performance. Our paper mainly makes the following contributions:

- **DecFus Paradigm:** To our best knowledge, DecFus is the first DFL paradigm that unifies layer-wise exchange and averaging with theoretical convergence guarantees, which eliminates the assumption of doubly stochastic property of aggregation matrix. It fuses layer-wise exchange for exploration and layer-wise averaging for exploitation, breaking the performance ceiling of vanilla model averaging.

- **Dynamic Exploration and Exploitation:** We use a decay cutoff to dynamically transition the aggregation process from an exploration-dominant exchange phase to an exploitation-dominant averaging phase, triggered by a stability detector that monitors training loss variance among connected neighbors.

- **Layer-wise Exchange and Averaging:** We propose a layer-wise aggregation strategy using pairwise cosine similarity and adaptive cutoff control to dynamically split layers into an exchange group for exploration and an averaging group for exploitation. This approach unifies layer averaging and exchange in a single aggregation framework.

- **Extensive Experiments:** We conduct extensive experiments on benchmark datasets under various settings, comparing our method with state-of-the-art CFL and DFL baselines. Simulation results verify the superiority of the proposed DecFus over existing baseline.

## 2. Preliminaries and Related Work

Consider a system with $N$ clients $\mathcal{N} = \{1, 2, \ldots, N\}$ connected via a decentralized graph $\mathcal{G}$, the DFL training process proceeds over $T$ rounds, with the following steps executed

sequentially in each round $t$:

- *Local training:* Each client $i$ trains its local model $\theta_i^t$ on its private dataset $\mathcal{X}_i$ in each round $t$, that is,

$$\theta_i^{t+1/2} = \theta_i^t - \eta \nabla \theta_i^t, \qquad (1)$$

where $\eta$ is the learning rate, $\nabla \theta_i^t$ is the gradient and $\theta_i^{t+1/2}$ denotes the local pre-aggregation model.

- *Consensus Aggregation:* Client $i$ exchanges its local trained model $\theta_i^{t+1/2}$ with its connected neighbors, and updates its model via a consensus aggregation method:

$$\theta_i^{t+1} = \sum_{j \in \mathcal{G}_i} w_{ij} \theta_j^{t+\frac{1}{2}}, \qquad (2)$$

where $\mathcal{G}_i$ denotes the sub-graph centered on client $i$ (including client $i$ and its neighbors), $\theta_i^{t+1}$ is the post-aggregation model for client $i$ in round $t+1$.

The aggregation matrix $W = [w_{ij}] \in \mathbb{R}^{N \times N}$ associated with the graph $\mathcal{G}$ is doubly stochastic, which satisfies $w_{ij} = w_{ji} \in [0, 1), \forall i, j$ and $\sum_{j=1}^N w_{ij} = 1, \forall i$ to ensure model convergence (Liu & Ding, 2021a; Liu et al., 2025a). To facilitate the subsequent discussion, a model $\theta_i$ is defined as a composition of multiple layers, represented as:

$$\theta_i = [L_{i,1}, L_{i,2}, \ldots, L_{i,r}, \ldots, L_{i,R}], \qquad (3)$$

where $L_{i,r}$ denotes the $r^{th}$ layer in model $\theta_i$, and $R$ is the total number of layers.

### 2.1. Related Work

Averaging-based aggregation underpins both CFL and DFL (McMahan et al., 2017), but its effectiveness diminishes in practical settings, particularly for clients with non-IID data. This issue is more pronounced in DFL, where sparse communication topology slows information propagation and amplifies local bias (Sun et al., 2023).

Many works have attempted to mitigate this issue by augmenting the averaging paradigm with various corrective mechanisms. In the CFL setting, methods like FedProx (Li et al., 2020) introduce a proximal term to constrain model drift, while CluSamp (Fraboni et al., 2021) and Fed-ExP (Jhunjhunwala et al., 2023) refine client selection and server-side updates, respectively. Similar strategies appear in DFL, where approaches like DisPFL (Dai et al., 2022) and DFedPGP (Liu et al., 2024) use personalized masks or partial gradient pushes to manage heterogeneity. Although effective to some extent, these approaches remain confined to averaging-based updates, making them prone to convergence toward suboptimal local minima in the complex, non-convex loss landscapes of deep networks.

Recognizing the limitations of averaging, recent research has begun to investigate the geometry of the loss landscape

itself. The work (Shi et al., 2023) aims to find "flatter" minima by employing sharpness-aware minimization in DFL. In CFL, FedMR (Hu et al., 2024) promotes broader exploration by replacing parameter averaging with parameter recombination. However, this unconstrained exploration often causes unstable oscillations around high-quality minima in later training stages, hindering deep and stable convergence.

These developments reveal a key limitation in current approaches: the lack of a unified mechanism to balance exploration and exploitation in the model aggregation process. To address this gap, we propose DecFus, a decentralized framework that unifies layer exchange and averaging into a single, dynamic layer-wise aggregation strategy. Different from model merging that operates at the whole-model level, DecFus performs layer-wise fusion, where each layer can be selectively exchanged or averaged.

## 3. Motivation

The potential of combining layer exchange for exploration and layer averaging for exploitation has been established in our previous discussion, but how to effectively orchestrate this fusion remains unclear. This raises the following two key questions in designing DecFus.

- How to balance layer exchange and averaging throughout training process?
- How to realize effective layer exchange and averaging at each training round?

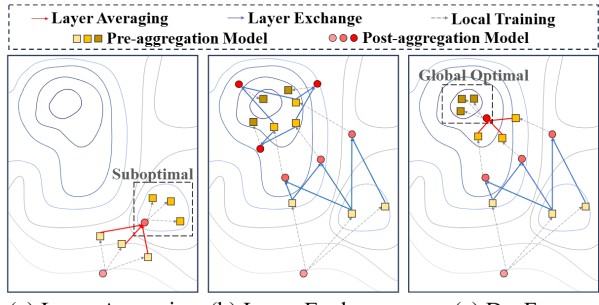

*Figure 1.* Training processes of various aggregation methods on the same loss landscape.

For the first question, we draw inspiration for an exploration-first and exploitation-later strategy from metaheuristic optimization methods (Ting et al., 2024) and learning rate decay in model training (Loshchilov & Hutter, 2017). We observe that averaging-based methods (Figure 1(a)) ensuring stable convergence but quickly get trapped in local minima (Hu et al., 2024), while exchange-based methods (Figure 1(b)) preserve diversity to escape poor local minima but often cause unstable oscillations in later stages, failing to exploit promising regions effectively.

To validate the above intuition, we measure the aggregation

shift, which quantifies the average distance of client model parameters before and after aggregation across different aggregation strategies. Formally, at communication round $t$, the aggregation shift is defined as

$$S^t = \frac{1}{N} \sum_{i=1}^{N} \left\| \theta_i^{t+1} - \theta_i^{t+\frac{1}{2}} \right\|_2. \qquad (4)$$

As shown in Figure 2, layer averaging exhibits low aggregation shift, ensuring stability but limiting exploration, whereas layer exchange shows high aggregation shift, promoting exploration in loss landscape. However, the aggregation shift still plateaus at a non-negligible level even in late stages, indicating instability risk, although the shift does decrease over training indicating that the method exhibits a convergent trend. DecFus mitigates these drawbacks by designing a dynamic cutoff, which starts with an exchange-dominant phase to escape poor optima, then gradually transitions to an averaging-dominant phase (Figure 1(c)) to refine the solution and reducing late-stage instability risk, balancing early exploration with later exploitation.

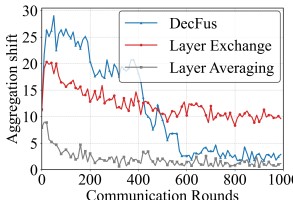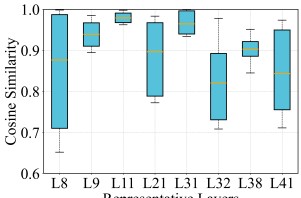

*Figure 2.* Aggregation shift of various methods.

*Figure 3.* Pairwise layer similarity distribution.

For the second question, we measure pairwise layer similarity among connected neighbors with a randomly sampled subset of layers (Figure 3). It is evident that the pairwise layer similarity vary significantly, where certain layers exhibit consistent similarity across clients, while others remain highly divergent. Therefore, uniform or random grouping strategies for implementing exchange and averaging fail to capture the varying degrees of layer-wise disagreement, thereby diminishing the effectiveness of both approaches. This observation motivates our similarity-based layer partitioning strategy in DecFus, that is, layers with low similarity are selectively exchanged to enhance exploration, while high-similarity layers are averaged to promote exploitation for unified exploration and exploitation.

## 4. Methodology

This section introduces the proposed DecFus framework and its communication-efficient extension, EDecFus.

### 4.1. Overview of DecFus

DecFus aims to construct a unified layer-wise fusion framework that dynamically performs layer-level exchange or

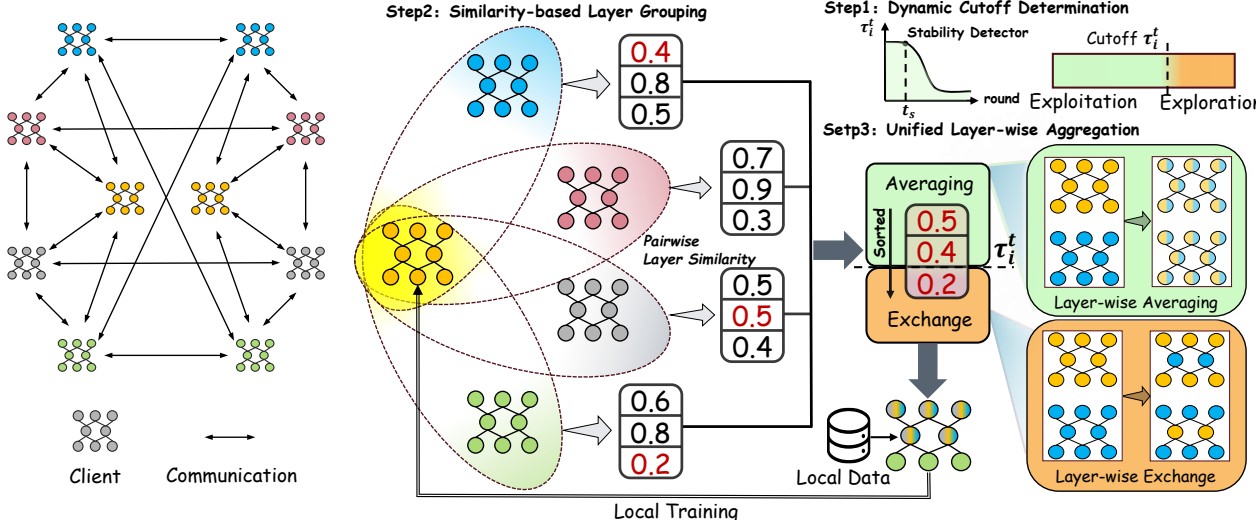

(a) Decentralized Communication topology      (b) Schematic diagram of DecFus

*Figure 4.* Framework of DecFus. (a) is an example of a decentralized communication topology employed in our experiments; (b) illustrates the overall three steps of the proposed DecFus.

averaging over different training stages. It consists of three components: a Dynamic Cutoff Determination module to decide group proportions, a Similarity-based Layer Grouping method to partition layers, and a Unified Layer-wise Aggregation mechanism to seamlessly fuse layer exchange and averaging. As shown in Figure 4, after local training to obtain intermediate model $\theta_i^{t+1/2}$, each client $i$ executes:

- *Dynamic Cutoff Determination*: Each client dynamically compute a cutoff $\tau_i^t$ to determine the group sizes of layer exchange and averaging. This cutoff starts from a value that encourages layer exchange toward a value that favors averaging, triggered by the training loss variance among connected neighbors.

- *Similarity-based layer grouping*: Following cutoff determination, client $i$ computes the pairwise cosine similarity of layers to obtain the layer similarity score $\sigma_{i,r}$. Using the dynamic cutoff $\tau_i^t$ over the set of layer similarity scores $\{\sigma_{i,r}\}_{r=1}^R$, client $i$ assigns layers with high similarity scores to the averaging group and those with low similarity scores to the exchange group.

- *Unified Layer-wise Aggregation*: Each group is aggregated using a specific strategy: layers in the averaging group perform weighted averaging, while those in the exchange group undergo probabilistic exchange. These strategies are seamlessly integrated into a unified layer-wise aggregation formula.

### 4.2. Dynamic Cutoff Determination (DCD)

A dynamic cutoff, functioning as a percentile, partitions all layers into two groups: exchange group and averaging group, with higher values indicating larger exchange group and smaller averaging group. This cutoff follows an adaptive

transition strategy triggered upon stabilization detection, gradually shifting aggregation from exploration-dominant exchange phase to exploitation-dominant averaging phase.

#### 4.2.1. NEIGHBORHOOD STABILITY DETECTOR

This design is inspired by the observation that layer exchange causes high, fluctuating training loss variance among neighbors in early rounds, which stabilizes as clients converge toward a shared basin. To capture this dynamic, we introduce a stability score to quantify neighborhood model convergence.

Specifically, client $i$ computes the neighborhood training loss variance $V_{\mathcal{G}_i}^t = \mathrm{var}(\mathcal{L}_j^{t-1}, j \in \mathcal{G}_i)$ in each round $t$, where $\mathcal{L}_i^{t-1}$ denotes the training loss in round $t-1$, and $\mathrm{var}(\cdot)$ is the variance. A sliding window of the past $w$ variances is then used to compute the stability score $\delta_i^t$:

$$\delta_i^t = \frac{\mathrm{std}(V_{\mathcal{G}_i}^\xi, \xi \in [t-w:t] \cap \mathbb{Z})}{\frac{1}{|\mathcal{T}_r|}\sum_{t \in \mathcal{T}_r \cap \mathbb{Z}} V_{\mathcal{G}_i}^t}, \qquad (5)$$

where $\mathcal{T}_r$ denotes predefined interval of early training rounds, and $\mathrm{std}(\cdot)$ is the standard deviation. Once the stability score $\delta_i^t$ drops below a predefined stability threshold $\epsilon_s$, the corresponding round is recorded as $t_s$ to initiate the cutoff transition process.

#### 4.2.2. DYNAMIC CUTOFF STRATEGY

The cutoff $\tau_i^t \in (0, 1)$ evolves over time:

$$\tau_i^t = \begin{cases} \tau_U, & \text{if } t < t_s \\ \tau_U - S\left(\frac{t-t_s}{\kappa T}\right)(\tau_U - \tau_L), & \text{if } t \geq t_s \end{cases}, \qquad (6)$$

where $S(x) = \frac{1}{1+e^{-\alpha(x-\beta)}}$ is the Sigmoid function parameterized by $\alpha$ and $\beta$, and $\tau_U$ and $\tau_L$ define the upper and lower bounds of the cutoff. The term $\frac{t-t_s}{\kappa T}$ measures the normalized training progress within the transition phase, where $\kappa$ specifies the fraction of total training rounds $T$ over which the cutoff transition is primarily completed.

## 4.3. Similarity-based Layer Grouping (SLG)

Each client computes a layer similarity score $\sigma_{i,r}$ for each layer $r \in R$ between its own layer and those of its neighbors. The set of layers $\{L_r\}_{r=1}^R$ is then partitioned based on these scores using a dynamic cutoff $\tau_i^t$: layers with higher scores are assigned to the averaging group, while those with lower scores are assigned to the exchange group.

Specifically, for each layer $r$, client $i$ computes the pairwise cosine similarity between its own layer vector $L_{i,r}$ and that of each connected neighbor $j$:

$$CS(L_{i,r}, L_{j,r}) = \frac{L_{i,r} \cdot L_{j,r}}{\|L_{i,r}\|\|L_{j,r}\|}, r \in R, \qquad (7)$$

where $CS$ denotes cosine similarity, and the layer similarity score $\sigma_{i,r}$ is defined as the minimum similarity of layer $r$ between client $i$ and its neighbors:

$$\sigma_{i,r} = \min_j CS(L_{i,r}, L_{j,r}), j \in \{\mathcal{G}_i \setminus i\}, \qquad (8)$$

where $\mathcal{G}_i \setminus i$ denote the subgraph excluding client $i$ itself.

Notably, the computational complexity of cosine similarity per round is $\mathcal{O}\left(C_{|\mathcal{G}_i|}^2 \cdot \sum_{r=1}^R d_r\right)$, which is relatively negligible compared to the complexity of local training, given by $\mathcal{O}\left(E \cdot \frac{|\mathcal{X}_i|}{B} \cdot \sum_{r=1}^R d_r\right)$. Here, $|\mathcal{G}_i|$ is client numbers in subgraph $\mathcal{G}_i$, $d_r$ is parameter numbers in layer $r$.

After determining the layer similarity scores $\{\sigma_{i,r}\}_{r=1}^R$, client $i$ partitions its model layers into two groups for different strategies. Specifically, the set of scores $\{\sigma_{i,r}\}_{r=1}^R$ is first sorted, and a cutoff value $\sigma_{i,c}$ is determined at the $\tau_i^t$-th quantile. Each layer $r$ is then assigned as follows:

- **Averaging Group** $\mathcal{P}_i^t = \{r \in \{1, \ldots, R\} \mid \sigma_{i,r} > \sigma_{i,c}\}$: Contains all layers where $\sigma_{i,r} > \sigma_{i,c}$, which performs layer averaging to refine the current solution.
- **Exchange Group** $\mathcal{Q}_i^t = \{r \in \{1, \ldots, R\} \mid \sigma_{i,r} \leq \sigma_{i,c}\}$: Contains all layers where $\sigma_{i,r} \leq \sigma_{i,c}$, which employs layer exchange to encourage exploration.

## 4.4. Unified Layer-wise Aggregation

After partitioning all layers into two groups of exchange and averaging, client $i$ applies various strategies to each group.

**Layer-wise Averaging (LA)** For layers in averaging group, client $i$ performs weighted average to update these layers

for exploitation, that is

$$L_{i,r}^{t+1} = v_{ij,r} \sum_{j \in \mathcal{G}_i} L_{j,r}^{t+\frac{1}{2}}, r \in \mathcal{P}_i^t, \qquad (9)$$

where $v_{ij,r} \in [0, 1)$ is the aggregation weight of layer $r$.

**Layer-wise Exchange (LE)** For layers in $\mathcal{Q}_i^t$, client $i$ performs a probabilistic layer exchange to update these layers for exploration, that is:

$$L_{i,r}^{t+1} = \begin{cases} L_{j^*,r}^{t+\frac{1}{2}}, & \text{with probability } \gamma \\ L_{k,r}^{t+\frac{1}{2}}, & \text{otherwise}, k \sim \{\mathcal{G}_i \setminus j^*\} \end{cases}, r \in \mathcal{Q}_i^t, \qquad (10)$$

where $j^*$ is the most dissimilar neighbor with minimum layer similarity score $\sigma_{i,r}$, and $k$ is the randomly chosen clients from remaining neighbors. $\gamma \in [0, 1]$ is a hyperparameter to control the layer exchange behavior.

Although layer averaging and exchange appear distinct, we propose the Unified Layer-wise Aggregation to unify these two methods in a single aggregation framework, that is

$$L_{i,r}^{t+1} = \sum_{j \in \mathcal{G}_i} v_{ij,r} L_{j,r}^{t+\frac{1}{2}}, \qquad (11)$$

$$v_{ij,r} = \begin{cases} w_{ij}, & \text{if } r \in \mathcal{P}_i^t \\ 0 \text{ or } 1, & \text{if } r \in \mathcal{Q}_i^t \end{cases}, \qquad (12)$$

where $v_{ij,r} \in [0, 1]$ is the layer-wise aggregation weight, and it satisfies $\sum_{j \in \mathcal{G}_i} v_{ij,r} = 1$ to ensure convergence.

Compared to the standard consensus aggregation (2) in DFL, the improvements of DecFus are two-fold. On the one hand, DecFus refines the consensus aggregation process by shifting from a model-level approach to a layer-level decomposition. This is achieved by transforming the aggregation weight from a model-level scalar $w_{ij}$ to a layer-level vector $v_{ij} = [v_{ij,1}, \ldots, v_{ij,R}]$, and correspondingly, from the aggregation matrix $W = [w_{ij}] \in \mathbb{R}^{N \times N}$ to $V = [v_{ij,r}] \in \mathbb{R}^{N \times N \times R}$. On the other hand, DecFus enables both layer averaging and exchange by extending the aggregation weight domain from a semi-open interval $[0, 1)$ to a closed interval $[0, 1]$, thereby unifying averaging and exchange in aggregation process. Through these enhancements, DecFus not only preserves the core advantages of DFL but also extends its performance potential.

DecFus unifies layer-wise exchange and averaging by utilizing a weight vector $v_i^r = [v_{ij,r}]_{j \in \mathcal{G}_i}$, where the choice of $v_i^r$ determines the operation performed. Specifically,

- **Layer-wise Averaging:** By defining $v_i^r$ as a probability vector with $v_{ij,r} \in [0, 1]$ and $\sum_{j \in \mathcal{G}_i} v_{ij,r} = 1$.
- **Layer-wise Exchange:** By defining $v_i^r$ as a one-hot vector with only one entry being 1 and others being 0.

**Algorithm 1** DecFus

1: **Input:** Client set $\mathcal{N}$, communication topology $\mathcal{G}$, datasets $\{\mathcal{X}_i\}_{i \in \mathcal{N}}$, total training rounds $T$, stability threshold $\epsilon_s$.
2: **Output:** Optimal models $\theta_i^T$ for client $i \in \mathcal{N}$.
3: **Initialization:** Model $\theta_i^0 = [L_{i,1}^0, ..., L_{i,r}^0, ..., L_{i,R}^0]$ for client $i \in \mathcal{N}$.
4: **for** $t = 1$ **to** $T$ **do**
5:     **for** each client $i \in \mathcal{N}$ **in parallel do**
6:         $\theta_i^{t+\frac{1}{2}} = [L_{i,1}^{t+\frac{1}{2}}, \dots, L_{i,R}^{t+\frac{1}{2}}] \leftarrow$ Train on $\mathcal{X}_i$ by (1).
7:     **end for**
8:     **for** each client $i \in \mathcal{N}$ **in parallel do**
9:         $\tau_i^t \leftarrow \text{DCD}(\epsilon_s, \mathcal{L}_j^{t-1}, j \in \{\mathcal{G}_i\})$ by (5),(6).
10:         **for** layer $r = 1$ **to** $R$ **do**
11:             $CS(L_{i,r}, L_{j,r}), j \in \{\mathcal{G}_i \setminus i\}$ by (7).
12:             $\sigma_{i,r} \leftarrow \min_j CS(L_{i,r}, L_{j,r}), j \in \{\mathcal{G}_i \setminus i\}$.
13:         **end for**
14:         $\mathcal{P}_i^t, \mathcal{Q}_i^t \leftarrow \text{Group}\{1, 2, ..., R\}$ by $\tau_i^t, \{\sigma_{i,r}\}_{r=1}^R$.
15:         **for** $r = 1$ **to** $R$ **do**
16:             **if** $r \in \mathcal{P}_i^t$ **then**
17:                 $L_{i,r}^{t+1} \leftarrow \text{LA}(L_{j,r}^{t+\frac{1}{2}}, j \in \{\mathcal{G}_i\})$ by (9).
18:             **end if**
19:             **if** $r \in \mathcal{Q}_i^t$ **then**
20:                 $L_{i,r}^{t+1} \leftarrow \text{LE}(L_{j,r}^{t+\frac{1}{2}}, j \in \{\mathcal{G}_i\})$ by (10).
21:             **end if**
22:         **end for**
23:     **end for**
24: **end for**

Notably, DecFus degrades into DFL by assigning $v_{ij,r} = w_{ij}$ and restricting $v_{ij,r} \in [0, 1)$.

Based on the above three main components, we introduce DecFus as detailed in Algorithm 1, along with its communication-efficient extension, EDecFus, which is further elaborated in the Appendix A.2.

## 5. Convergence Analysis

This section provides the main idea of convergence analysis on DecFus algorithm. Following (Lian et al., 2017; Tang et al., 2018; Zeng et al., 2025) , we make the following Assumptions 5.1-5.3, which are commonly used in decentralized learning convergence analysis.

**Assumption 5.1** (Smoothness). Each local objective function $f_i(x)$ is $L$-smooth, i.e., $\|\nabla f_i(x) - \nabla f_i(y)\| \le L\|x - y\|, \forall i \in \mathcal{N}$.

**Assumption 5.2** (Unbiasedness). we assume an unbiased stochastic gradient $\nabla F_i(x, \xi_i)$ of the true gradient $\nabla f_i(x)$, i.e., $\mathbb{E}_{\xi_i \sim \mathcal{D}_i}[\nabla F_i(x, \xi_i)] = \nabla f_i(x), \forall i \in \mathcal{N}$.

**Assumption 5.3** (Bounded Variance). We assume a bounded average variance, i.e., $\mathbb{E}\|\nabla f_i(x) - \nabla f(x)\|^2 \le \sigma_g^2$ and $\mathbb{E}_{\xi_i \sim \mathcal{D}_i}\left[\|\nabla F_i(x, \xi_i) - \nabla f_i(x)\|^2\right] \le \sigma_l^2$.

**Assumption 5.4** (Impermanent Layer Exchange). For each layer of all clients, We assume that the layer exchange probability $q = \frac{1}{T}\sum_{t=1}^T \mathbf{1}(v_{ij,r}^t = 1), \forall i, j, r$, this leads to $\exists t \in [1, 2, \cdots, T]$, we have $v_{ij,r}^t \ne 1, \forall i, j, r$.

**Remark 1**: Assumption 5.4 is easy to satisfy in DecFus, since the dynamic cutoff $\tau_i^t$ gradually shifts training from layer exchange to layer averaging, ensuring any layer participated in averaging in at least one training round. Furthermore, the hyperparameter $\gamma$ in (10) ensure that there will no certain layer being exchanged all the time.

**Remark 2**: Most decentralized learning methods assume the aggregation matrix in (2) is a doubly stochastic matrix to satisfy the following Proposition 5.5 for convergence analysis. In this work, Assumption 5.4 is a more general assumption compared to doubly stochastic property, as it exists $\lambda_2(\mathbb{E}[V_r^{t T} V_r^t]) = 1$. Therefore, existing convergence analysis is not applicable to DecFus. Notably, when $q = 0$, we have $V_r^t$ degrade into a doubly stochastic matrix and satisfy Proposition 5.5.

**Proposition 5.5.** *The doubly stochastic matrix satisfies:*

$$\|\frac{1_N}{N} - \prod_{t=1}^T W^t e_i\| \le \prod_{t=1}^T \rho^t, \quad \forall i \in \mathcal{N}, \forall t \in [1, 2, \cdots, T], \tag{13}$$

*where* $\rho^t \triangleq |\lambda_2(\mathbb{E}[W^{t T} W^t])| \in [0, 1)$, $1_N \in \mathbb{R}^{N \times 1}$ *is a column vector with all element being 1, $e_i \in \mathbb{R}^{N \times 1}$ is standard basis with the $i-th$ element being 1, and other elements being 0.*

Obviously, the matrix $V_r = [v_{ij,r}] \in \mathbb{R}^{N \times N}$ does not satisfy Proposition 5.5. We innovatively introduce Lemma 5.6 based on Assumption 5.4 to replace Proposition 5.5 for the convergence analysis on DecFus. The proof of Lemma 5.6 refers to Appendix A.1.2.

**Lemma 5.6.** *There exists $1 \le m \le N^2$, $\frac{1}{N^2 RT} \le \epsilon \le \frac{1}{N^2}$, such that:*

$$\max_r \|\frac{1_N}{N} - \prod_{t=1}^T V_r^t e_i\| \le \prod_{t=1}^T \widetilde{\rho}^t, \tag{14}$$

*where* $\widetilde{\rho}^t = \max_r(\rho_r^t)\left(1 - q(1 - N\epsilon)^2\right)^m \in [0, 1)$, $\rho_r^t \triangleq |\lambda_2(\mathbb{E}[V_r^{t T} V_r^t])| \in [0, 1]$.

**Remark 3**. When $q > 0$, layer exchange occurs in DECFUS, causing $V_r^t$ to deviate from being a doubly stochastic matrix, and thus violating Proposition 5.5. To address this, we introduce an independent Markov process to prevent the absorbing state of layer exchange. This mechanism ensures that the spectral radius of the cumulative product of $V_r^t$ remains strictly less than 1, i.e., $\widetilde{\rho}^t < 1$. In Lemma 5.6, $m$

denotes the interval between successive appearances of 1 in the Markov process, while $\epsilon$ is an infinitesimal that governs the recursion of the Markov process.

**Remark 4**. When $q = 0$, only layer averaging occurs and no layer exchange takes place. In this case, $V_r^t$ degenerates to a doubly stochastic matrix, and Proposition 5.5 holds.

**Lemma 5.7.** *Let $\theta_i^{t+1/2}$ and $\theta_i^t$ be the client's model before and after the layer-wise aggregation in DecFus, we have*

$$\frac{1}{N}\sum_{i=1}^{N}\|\theta_i^t\| = \frac{1}{N}\sum_{i=1}^{N}\|\theta_i^{t+1/2}\|. \tag{15}$$

Based on Lemma 5.6 and 5.7, we provide the detailed proof of Theorem 5.8 in the Appendix A.1.

**Theorem 5.8.** *Given that Assumptions 5.1–5.3 and 5.4 hold, and the step size $\eta \leq \frac{1}{2L+\sigma_g\sqrt{\frac{T}{N}}}$ such that the decentralized convergence of DecFus can be expressed as follows:*

$$\frac{\mathbb{E}\left[\sum_{t=0}^{T-1}\sum_{i=1}^{N}\left\|\frac{\sum_{i'=1}^{N}\theta_{i'}^t}{N} - \theta_i^t\right\|^2\right]}{TN}$$
$$\leq \frac{18\eta^2 N}{(1-\sqrt{\widetilde{\rho}})^2\widetilde{D_2}}\left(\frac{f(0)-f^*}{\eta T} + \frac{\eta L\sigma_l^2}{2N\widetilde{D_1}}\right) + \frac{2\eta^2 N\sigma_l^2}{(1-\widetilde{\rho})\widetilde{D_2}}$$
$$+ \frac{18\eta^2 N\sigma_g^2}{(1-\sqrt{\widetilde{\rho}})^2\widetilde{D_2}} + \frac{\eta^2 L^2 N}{\widetilde{D_1}\widetilde{D_2}}\left(\frac{\sigma_l^2}{1-\widetilde{\rho}} + \frac{9\sigma_g^2}{(1-\sqrt{\widetilde{\rho}})^2}\right),$$
$$\tag{16}$$
*where $\widetilde{D_1} = \frac{1}{2} - \frac{9\sigma_l^2 L^2 n}{(1-\sqrt{\widetilde{\rho}})^2\widetilde{D_2}}$, $\widetilde{D_2} = 1 - \frac{18\sigma_l^2}{(1-\sqrt{\widetilde{\rho}})^2 NL^2}$, $\widetilde{\rho} = \max_t \widetilde{\rho}^t$.*

**Remark 5**: Based on Lemma 5.6, we still have $\widetilde{\rho} \in [0, 1)$, which shows that DecFus has similar convergence rate compared to existing decentralized learning methods in (Lian et al., 2017; Tang et al., 2018).

## 6. Experiments

### 6.1. Experimental Setup

To evaluate the effectiveness of DecFus and EDecFus, we conduct experiments on different settings under the decentralized communication topology illustrated in Figure 4(a). More detailed experimental settings, parameters and experiments can be found in Appendix A.3, including computation overhead comparisons, communication-cost and performance under different decentralized topologies.

**Dataset and Model Settings.** We investigate the performance on three datasets: CIFAR-10/100 (Krizhevsky & Hinton, 2009) and SVHN (Netzer et al., 2011). Additionally, we use ResNet-50 (He et al., 2016) and VGG-16 (Simonyan & Zisserman, 2015) to evaluate our proposed method. To

control the degree of data heterogeneity across clients, we adopt the Dirichlet distribution $Dir(\alpha)$ (Hsu et al., 2019). The detail is provided in the Appendix A.3.1.

**Hyper-Parameters.** The key hyperparameters used in our experiments include the stability threshold $\epsilon_s = 0.35$, a Sigmoid hyperparameter $\alpha = 20$, $\beta = 0.4$ and the expected cutoff $\kappa = 0.5$ and $\gamma = 0.4$ which control the layer exchange behavior. The other parameters are provided in the Appendix A.3.2. It is worth noting that all key hyperparameters are kept fixed across all datasets and data distribution, demonstrating the robustness of the proposed algorithm.

**Baseline Method Settings.** To comprehensively evaluate the effectiveness of our proposed algorithms, we conduct comparisons against a range of representative baselines, including both decentralized extensions of CFL methods and DFL strategies.

Specifically, we include the following decentralized extensions of state-of-the-art CFL methods:

- **FedProx** (Li et al., 2020): A proximal-regularized method introduces a control variable to mitigate local model drift, whose decentralized variant uses the previous round neighborhood averaged model as the proximal reference.
- **FedExP** (Jhunjhunwala et al., 2023): An extrapolated method that adaptively scales the aggregation step size, whose decentralized extension computes scaling from neighborhood updates.
- **FedMR** (Hu et al., 2024): A method that replaces model averaging with parameter recombination, whose decentralized extension performs structured parameter recombination over neighborhood clients.

For DFL-based comparisons, we include the following representative decentralized methods:

- **DFLAvg**: The canonical decentralized federated averaging algorithm as the fundamental baseline for DFL.
- **DFedSAM** (Caldarola et al., 2022): A stochastic perturbation-based method that improves generalization by encouraging exploration toward flatter minima in the loss landscape.
- **DisPFL** (Dai et al., 2022): A personalized method that employs sparse parameter masking and selective model updates to adapt heterogeneous data distributions.

### 6.2. Performance Comparison

Table 1 presents the quantitative results of DecFus and EDecFus, along with comparisons to state-of-the-art methods across three datasets and two model architectures. DecFus outperforms the baselines under various levels of non-IID settings in most cases and EDecFus achieves comparable performance with reduced communication overhead. The

*Table 1.* Test accuracy comparison on various models and datasets for both non-IID and IID scenarios.

| Model | Dataset | Heter. Set. | Test Accuracy (%) | | | | | | | |
|---|---|---|---|---|---|---|---|---|---|---|
| | | | DFLAvg | DFLProx | DFLExP | DisPFL | DFedSAM | DFLMR | DecFus | EDecFus |
| ResNet-50 | Cifar-10 | 0.1 | 32.72±0.71 | 32.80±0.57 | 31.91±0.30 | 35.63±0.28 | 30.90±0.32 | 32.76±0.34 | **36.60±0.40** | 33.14±0.26 |
| | | 0.5 | 68.84±0.45 | 68.29±0.26 | 68.33±0.07 | 69.22±0.23 | 65.47±0.25 | 68.51±0.41 | **73.39±0.29** | 71.02±0.16 |
| | | 1.0 | 69.82±1.02 | 69.97±0.23 | 68.22±0.18 | 73.09±0.02 | 68.50±0.23 | 72.04±0.86 | **74.96±0.26** | 72.34±0.14 |
| | | IID | 74.07±0.23 | 73.02±0.17 | 73.58±0.19 | 61.45±1.73 | 74.60±0.38 | 76.02±0.12 | **78.24±0.12** | 76.93±0.36 |
| | Cifar-100 | 0.1 | 30.55±1.69 | 29.36±1.31 | 31.10±2.04 | 27.67±1.25 | 28.71±1.12 | 28.83±0.72 | **35.76±0.89** | 31.58±0.87 |
| | | 0.5 | 48.42±0.53 | 48.80±0.61 | 48.87±1.22 | 53.21±1.15 | 48.72±0.41 | 50.99±0.40 | **53.86±0.42** | 51.71±0.37 |
| | | 1.0 | 49.17±0.31 | 49.27±0.24 | 48.88±0.34 | 54.54±0.92 | 50.93±0.57 | 52.10±0.33 | **56.04±0.11** | 52.76±0.11 |
| | | IID | 52.77±0.62 | 52.27±0.31 | 53.28±0.20 | 57.39±0.64 | 54.13±0.37 | 55.91±0.43 | **58.05±0.43** | 56.17±0.31 |
| | SVHN | 0.1 | 53.32±0.59 | 54.10±0.78 | 51.60±0.97 | 52.24±1.73 | 50.51±0.51 | 54.27±0.61 | **57.83±0.85** | 56.35±0.79 |
| | | 0.5 | 82.48±0.09 | 83.03±0.13 | 82.88±0.85 | 80.93±0.21 | 82.26±1.30 | 84.90±0.07 | **86.33±0.22** | 85.17±0.03 |
| | | 1.0 | 83.45±3.21 | 85.90±0.60 | 86.52±0.96 | 81.85±1.68 | 84.99±2.54 | 84.33±2.68 | **89.31±0.68** | 89.08±0.42 |
| | | IID | 91.38±0.13 | 91.48±0.16 | 91.20±0.08 | 92.10±0.10 | 91.98±1.12 | 92.09±0.03 | **93.43±0.04** | 92.17±0.04 |
| VGG16 | Cifar-10 | 0.1 | 53.64±6.31 | **54.36±4.50** | 53.67±1.06 | 42.07±0.13 | 51.85±1.09 | 42.85±0.90 | 54.14±2.44 | 52.98±3.27 |
| | | 0.5 | 80.70±0.27 | 80.57±0.29 | 80.12±0.14 | 79.91±0.52 | 81.42±0.27 | 80.41±1.26 | **83.91±0.27** | 83.03±0.87 |
| | | 1.0 | 80.89±0.13 | 81.10±0.23 | 80.99±0.16 | 81.47±1.20 | 81.52±0.57 | 80.43±0.60 | **84.23±0.11** | 83.34±0.36 |
| | | IID | 82.97±0.18 | 83.07±0.15 | 83.49±0.18 | 83.61±0.21 | 83.07±0.52 | 84.05±0.48 | **86.28±0.23** | 85.03±0.33 |
| | Cifar-100 | 0.1 | 42.78±3.09 | 43.15±2.99 | 44.73±1.70 | 31.88±0.20 | 43.36±3.74 | 36.01±0.72 | **44.91±3.68** | 42.55±1.56 |
| | | 0.5 | 59.71±0.08 | 59.95±0.40 | 59.41±0.09 | 58.55±0.45 | 60.81±0.26 | 60.29±0.14 | **65.14±0.15** | 63.87±0.14 |
| | | 1.0 | 60.22±0.08 | 60.36±0.12 | 60.54±0.23 | 58.25±0.10 | 61.56±0.32 | 62.37±0.12 | **66.39±0.08** | 64.31±0.04 |
| | | IID | 62.10±0.13 | 62.04±0.10 | 62.49±0.10 | 55.80±0.44 | 62.26±0.11 | 64.91±0.32 | **67.52±0.17** | 65.41±0.44 |
| | SVHN | 0.1 | 75.02±0.49 | **75.65±0.88** | 73.64±1.06 | 54.61±2.58 | 71.54±1.22 | 70.25±0.41 | 72.84±1.57 | 72.63±1.50 |
| | | 0.5 | 92.20±0.07 | 92.02±0.07 | 91.75±0.02 | 83.34±0.31 | 91.91±0.21 | 90.60±0.17 | 91.60±0.15 | **92.36±0.11** |
| | | 1.0 | 92.95±0.41 | 93.58±1.30 | 93.04±1.89 | 88.61±1.54 | 92.28±0.61 | 89.79±1.78 | **93.97±1.26** | 92.99±1.96 |
| | | IID | 94.22±0.02 | 94.28±0.04 | 94.20±0.05 | 94.63±0.10 | 94.37±0.08 | 94.50±0.09 | 94.79±0.11 | **94.85±0.06** |

*(a)* $\alpha = 0.1$    *(b)* $\alpha = 0.5$    *(c)* $\alpha = 1.0$    *(d)* IID

*Figure 5.* Learning curves of various DFL methods using ResNet-50 on CIFAR-100 dataset.

performance improvements on the SVHN dataset are less pronounced compared to CIFAR-10 and CIFAR-100, primarily due to the simpler classification task of SVHN.

To compare the convergence speed, we also visualize the learning curves of six baselines and our proposed DecFus and EDecFus. Figure 5 shows that DecFus consistently outperforms all other methods under several settings.

### 6.3. Ablation Study

To verify the effectiveness of DecFus, we conduct ablation studies on DCD, similarity metric selection, and SLG.

**Dynamic Cutoff Determination (DCD).** To demonstrate the effectiveness of the proposed dynamic cutoff determination, we compare the following three variants:

- **Reversed Cutoff**: Replaces the decaying Sigmoid function with a rising one, gradually shifting the sys-

tem from an averaging-dominant phase to an exchange-dominant one.

- **High Cutoff**: Applies a fixed high threshold throughout training, consistently favoring exploration (exchange).
- **Low Cutoff**: Applies a fixed low threshold throughout training, consistently favoring exploitation (averaging).
- **DCD, Ours**: Employs our full dynamic threshold calculation, where the threshold decay is triggered by the Local Stability Calculator that monitors the relative variance of training losses.

As shown in Table 2, DecFus consistently outperforms the three variants under different datasets. Notably, the reversed cutoff perform poorly, as it enforces exploration in the late training stage affected model stabilization.

**Similarity Metric Selection.** To further evaluate the effect of different similarity metrics in SLG, we replace cosine

*Table 2.* Ablation of Dynamic Cutoff Determination on ResNet50.

| Method Setting | Reversed Cutoff | High Cutoff | Low Cutoff | DCD (Ours) |
|---|---|---|---|---|
| Cifar10 $\alpha$=0.1 | 32.31±1.03 | 31.99±0.64 | 32.18±0.66 | **36.60±0.40** |
| Cifar10 IID | 75.52±0.40 | 77.52±0.42 | 76.23±0.39 | **78.24±0.12** |
| Cifar100 $\alpha$=0.1 | 26.28±2.34 | 30.23±1.32 | 30.50±1.52 | **35.76±0.89** |
| Cifar100 IID | 55.26±0.16 | 57.12±0.27 | 54.34±0.59 | **58.05±0.43** |
| SVHN $\alpha$=0.1 | 45.85±1.94 | 52.68±1.45 | 47.99±1.09 | **57.83±0.85** |
| SVHN IID | 91.90±0.18 | 93.03±0.04 | 91.39±0.03 | **93.43±0.04** |

*Table 3.* Ablation of Similarity Metrics on ResNet50.

| Method Setting | L2 Distance | KL Divergence | Cosine Similarity |
|---|---|---|---|
| Cifar10 $\alpha$=0.1 | 35.96±0.88 | 33.33±1.50 | **36.60±0.40** |
| Cifar10 IID | **78.75±0.37** | 74.93±0.69 | 78.24±0.12 |
| Cifar100 $\alpha$=0.1 | 33.05±1.93 | 32.79±1.20 | **35.76±0.89** |
| Cifar100 IID | 57.07±0.22 | 55.47±0.74 | **58.05±0.43** |
| SVHN $\alpha$=0.1 | 56.98±1.10 | 53.58±2.46 | **57.83±0.85** |
| SVHN IID | 93.10±0.03 | 93.32±0.07 | **93.43±0.04** |

similarity with two alternative distance measures:

- **L2 Distance**: Computes the Euclidean distance between layer parameters, where smaller distances indicate higher similarity.
- **KL Divergence**: Measures the divergence between layer parameter distributions, where smaller values imply higher similarity.
- **Cosine Similarity (Ours)**: Computes the cosine similarity between layer vectors, emphasizing the angle between them rather than magnitude differences.

The results in Table 3 indicate that cosine similarity consistently outperforms L2 distance and KL divergence across various dataset settings.

**Similarity-based Layer Grouping (SLG).** To assess the impact of our similarity-based layer grouping, we further design the following three variants:

- **Random Grouping**: Randomly group all layers into exchange and averaging groups, ignoring similarity.
- **Inverse Grouping**: Uses an inverse strategy to ours, high-similarity layers join exchange group, while low-similarity layers join averaging group.
- **SLG(Ours)**: Layers with high pairwise similarity are assigned to averaging group, while low-similarity layers are assigned to exchange group.

The experimental results in Table 4 show that DecFus outperforms the other variants across multiple tasks, demonstrating the importance of similarity-based layer grouping.

*Table 4.* Ablation of Similarity-based Layer Grouping on VGG16.

| Method Setting | Random Grouping | Inverse Grouping | SLG (Ours) |
|---|---|---|---|
| Cifar10 $\alpha$=0.1 | 48.36±1.57 | 40.04±1.52 | **54.14±2.44** |
| Cifar10 IID | 85.49±0.22 | 82.61±0.45 | **86.28±0.23** |
| Cifar100 $\alpha$=0.1 | 44.40±2.80 | 31.62±2.22 | **44.91±3.68** |
| Cifar100 IID | 66.32±0.44 | 58.85±0.43 | **67.52±0.17** |
| SVHN $\alpha$=0.1 | 71.47±1.23 | 63.50±1.71 | **72.84±1.57** |
| SVHN IID | 94.63±0.06 | 93.95±0.27 | **94.79±0.11** |

*Table 5.* Parameter sensitivity analysis.

| Setting | $\epsilon_s$ | $\alpha$ | $\beta$ | $\kappa$ | $\gamma$ | Test Acc. (%) |
|---|---|---|---|---|---|---|
| 1 | 0.33 | 15 | 0.35 | 0.45 | 0.35 | 93.44±0.30 |
| 2 | 0.33 | 25 | 0.45 | 0.55 | 0.45 | 92.98±1.36 |
| 3 | 0.33 | 15 | 0.35 | 0.55 | 0.45 | 93.22±1.12 |
| 4 | 0.33 | 25 | 0.45 | 0.45 | 0.35 | 93.73±0.65 |
| 5 | 0.37 | 15 | 0.45 | 0.45 | 0.45 | 91.84±1.80 |
| 6 | 0.37 | 25 | 0.35 | 0.55 | 0.35 | 93.83±1.81 |
| 7 | 0.37 | 15 | 0.45 | 0.55 | 0.35 | 93.30±0.57 |
| 8 | 0.37 | 25 | 0.35 | 0.45 | 0.45 | 92.39±2.56 |
| **Default** | 0.35 | 20 | 0.40 | 0.50 | 0.40 | **93.97±1.26** |

## 6.4. Parameter Sensitivity Analysis

To further evaluate the parameter sensitivity of DecFus, we conducted a multi-factor orthogonal sensitivity study on SVHN with VGG16 under the non-IID setting of $\alpha = 1$.

As shown in Table 5, these results show that varying these hyperparameters has only a limited impact on the performance, while DecFus consistently outperforms most baselines. These hyperparameters mainly shape the transition from exploration-dominant exchange to exploitation-dominant averaging, without affecting the core mechanism. This indicates that DecFus does not rely on careful hyperparameter tuning.

## 7. Conclusion

In this paper, we propose Decentralized Layer-wise Fusion (DecFus), the first DFL framework to unify layer exchange and averaging, effectively breaking the performance ceiling of existing averaging-based DFL methods. DecFus introduces a dynamic cutoff and similarity-based layer grouping to structure layer-wise exchange and averaging for coordinated exploration–exploitation transition. Furthermore, we propose a unified layer-wise aggregation approach that integrates layer exchange and averaging within a single aggregation framework, and we theoretically establish the convergence without relying on the common doubly stochastic aggregation matrix assumption. Extensive experiments across diverse scenarios demonstrate that DecFus outperform existing state-of-the-art DFL methods.

## Acknowledgements

This work is supported by the National Natural Science Foundation of China (Grant No.62203309) and the Guangdong Basic and Applied Basic Research Foundation (Grant No. 2024A1515011333).

## Impact Statement

This paper presents research aimed at advancing the field of Machine Learning, particularly in decentralized and federated learning systems. The proposed methods focus on improving training efficiency, robustness, and scalability in distributed learning environments, which may support the development of more reliable and efficient machine learning infrastructures.

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

# A. Appendix

This Appendix offers a comprehensive overview of the proof of convergence, extended method framework, experimental setup, results, and analyses that were omitted from the main paper due to space constraints. Specifically, we begin by proving the theorem. Then, we detail the design and implementation of the EDecFus framework. Next, we elaborate on the experimental setup, including experiment parameters. Finally, we present additional evaluation results across diverse communication topologies. The implementation is available at: https://github.com/Ublgod/DecFus-ICML2026.

## A.1. Proof of Theorem 5.8

This section focuses on the proof of Theorem 5.8. It begins with a brief introduction to the main idea of the proof, followed by the proof of Lemma 5.6, and concludes with the proof of Theorem 5.8, building upon the previous results.

### A.1.1. THE IDEA OF THE PROOF OF THEOREM

To prove Theorem 5.8, we first clarify the differences between DecFus and traditional decentralized federated learning convergence algorithms. In traditional decentralized algorithms, the weight matrix is denoted as $W^t \in \mathbb{R}^{N \times N \times R}$, where $N$ is the number of clients and $R$ is the number of model layers. Note that although $W^t$ contains $R$ matrices, these matrices are identical. Consequently, $W^t$ does not perform aggregation across different layers; this notation is introduced solely to maintain structural consistency with $V^t$. A more precise mathematical expression is given by:

$$
\begin{aligned}
W^t &= [W_1^t, W_2^t, \ldots, W_R^t] \in \mathbb{R}^{N \times N \times R}, \ W_r^t \in \mathbb{R}^{N \times N}, \ r \in \{1, 2, \ldots, R\}, \\
V^t &= [V_1^t, V_2^t, \ldots, V_R^t] \in \mathbb{R}^{N \times N \times R}, \ V_r^t \in \mathbb{R}^{N \times N}, \ r \in \{1, 2, \ldots, R\}.
\end{aligned}
\tag{17}
$$

In the following discussion, we use the weight matrix of a four-node ring topology to illustrate the differences between $W^t$ and $V^t$ in the DecFus algorithm. The topology of the four nodes is shown in Figure 6.

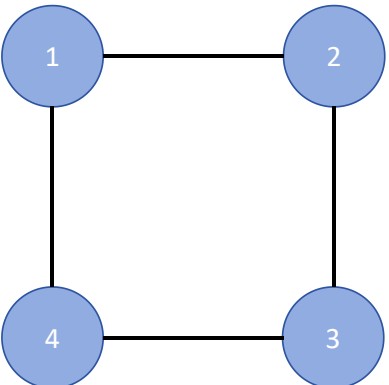

*Figure 6.* Four-node ring topology.

Under the four-node topology, a specific example of $W^t$ is as follows:

$$
\begin{aligned}
W^t &= \left[ W_1^t, \ldots, W_R^t \right] \\
&= \left[ \begin{bmatrix} \frac{1}{3} & \frac{1}{3} & 0 & \frac{1}{3} \\ \frac{1}{3} & \frac{1}{3} & \frac{1}{3} & 0 \\ 0 & \frac{1}{3} & \frac{1}{3} & \frac{1}{3} \\ \frac{1}{3} & 0 & \frac{1}{3} & \frac{1}{3} \end{bmatrix}, \ldots, \begin{bmatrix} \frac{1}{3} & \frac{1}{3} & 0 & \frac{1}{3} \\ \frac{1}{3} & \frac{1}{3} & \frac{1}{3} & 0 \\ 0 & \frac{1}{3} & \frac{1}{3} & \frac{1}{3} \\ \frac{1}{3} & 0 & \frac{1}{3} & \frac{1}{3} \end{bmatrix} \right].
\end{aligned}
\tag{18}
$$

Due to the properties of doubly stochastic matrices, both $W_1^t$ and $W_R^t$ satisfy $\lambda < 1$. This property implies that, during iterative aggregation, the product of these matrices will eventually converge to a stationary distribution. However, in the

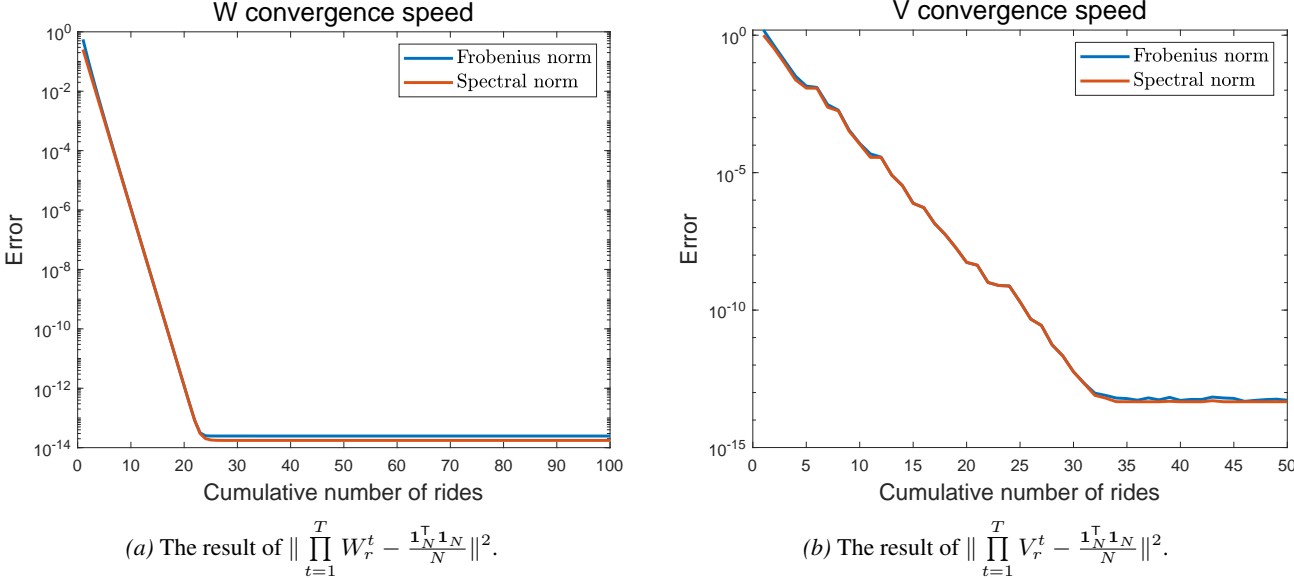

*(a)* The result of $\| \prod_{t=1}^{T} W_r^t - \frac{\mathbf{1}_N^\mathsf{T} \mathbf{1}_N}{N} \|^2$.

*(b)* The result of $\| \prod_{t=1}^{T} V_r^t - \frac{\mathbf{1}_N^\mathsf{T} \mathbf{1}_N}{N} \|^2$.

*Figure 7.* Comparison of the asymptotic convergence properties of $W^t$ and $V^t$.

DecFus algorithm, the exchange-layer matrix takes the following form:

$$
V^t = \left[ V_1^t, \ldots, V_R^t \right]
$$

$$
= \left[ \begin{bmatrix} 0 & 0 & 0 & 1 \\ 0 & \frac{1}{2} & \frac{1}{2} & 0 \\ 0 & \frac{1}{2} & \frac{1}{2} & 0 \\ 1 & 0 & 0 & 0 \end{bmatrix}, \ldots, \begin{bmatrix} \frac{1}{3} & \frac{1}{3} & 0 & \frac{1}{3} \\ \frac{1}{3} & \frac{1}{3} & \frac{1}{3} & 0 \\ 0 & \frac{1}{3} & \frac{1}{3} & \frac{1}{3} \\ \frac{1}{3} & 0 & \frac{1}{3} & \frac{1}{3} \end{bmatrix} \right]. \tag{19}
$$

This leads to two important implications: 1) The aggregation matrix of each layer must be analyzed individually, as the matrices are not identical; and 2) The $V_1$ matrix satisfies $\lambda = 1$. These properties pose a substantial challenge for convergence analysis. Classical convergence results require every matrix to satisfy $\lambda_2 < 1$, and it is evident that the self-multiplicative powers of $(V_1)^t$ do not readily converge to a stationary distribution. Furthermore, an additional difficulty arises because layer exchange does not occur continuously within a single layer. For instance, if two nodes in the DecFus process do not participate in any aggregation but only perform layer exchange with each other, the algorithm cannot be guaranteed to converge, as those nodes remain isolated from all other information.

To address this issue, the DecFus algorithm ensures that no two layers can aggregate exclusively with each other without also aggregating with other connected topological nodes. This design choice reinforces our confidence in the convergence guarantee of DecFus.

For this reason, the first step in establishing the convergence of DecFus is to determine whether the cumulative product of the $V$ matrices can converge to a stationary state. Before presenting the formal proof, we first conduct a simple numerical simulation to examine the convergence behavior of the $V^t$ matrix, followed by the corresponding proof strategy.

A simple numerical experiment, illustrated in Figure 7, compares the asymptotic convergence behavior of traditional centralized matrices with that of DecFus's weight matrices.

The simulation results indicate that, in most cases, DecFus achieves convergence performance that is nearly identical to traditional approaches, thereby reinforcing confidence in the subsequent convergence analysis and proof. The core challenge in establishing the convergence of DecFus is to demonstrate that the process in which two layers repeatedly exchange information cannot persist indefinitely.

A.1.2. PROOF OF LEMMA 5.6

Considering that the occurrence of layer exchange can be modeled as a Markov process, the key requirement is that mutual exchange between two layers does not enter an absorbing state, which enables a precise description of the process. To establish Lemma 5.6, we introduce Lemma A.1 as a preliminary result.

**Lemma A.1.** *Define the states of an external Markov random process $\mathcal{M} = \{1, \ldots, M\}$ as follows: for all $d \in \mathcal{M}$, there exist $P_{r,d}, Q_{r,d} \in \mathbb{R}^{N \times N}$. Let $V_{r,d} \in \mathbb{R}^{N \times N}$ denote the DecFus weight-exchange matrix, and let $A_{r,d} \in \mathbb{R}^{(N-1) \times (N-1)}$ denote the corresponding doubly stochastic matrix:*

$$V_{r,d} = P'_{r,d} \begin{bmatrix} 1 & 0 \\ 0 & A_{r,d} \end{bmatrix} Q_{r,d}. \tag{20}$$

*Since this formulation describes the layer-exchange process using a linear transformation, let the unique 1 be located at $P_{r,d}$ and $Q_{r,d}$. Define $V_d = V_{r,d} \odot \cdots \odot V_{1,d}$. There exists a finite integer $\kappa \geq 1$, a sequence $H \in \mathcal{M}^\kappa$, and a constant $\epsilon > 0$ such that, for all $\kappa = (\kappa^1, \ldots, \kappa^m)$, we have: $B_\kappa = \mathcal{V}_{\kappa^m} \circ \cdots \circ \mathcal{V}_{\kappa^1}$ is strictly positive, and each entry of $B_\kappa$ satisfies $(B_\kappa)_{ij} \geq \epsilon > 0$ for all $i, j \in \{1, \ldots, n\}$.*

**Remark A.1:** The symbol $\odot$ denotes the Hadamard product of matrices, and the symbol $\circ$ denotes the Hadamard product of sets. Here, $\circ$ additionally encodes the requirement that the elements of the sets share the same ordering.

**Remark A.2:** The Markov process with state space $\mathcal{M}$ in Lemma A.1 is introduced for the following reason. We assume that after two rounds of global aggregation, the model weight matrix $V^t$ changes. Between these two rounds, that is, during the transition from $V^t$ to $V^{t+1}$, the coordinates $(i, j, r)$ of the matrix entries equal to 1 evolve according to a sufficiently long Markov process. The main objective of the proof is to show that, regardless of the initial position of an entry equal to 1 at coordinates $(i, j, r)$, there is no absorbing state that prevents $(i, j, r)$ from changing throughout the entire DecFus iteration horizon $t = \{1, \ldots, T\}$.

*Proof.* Fix $\psi \in \{1, \ldots, N_r\}$, and define $S_\psi^r = \{d \in \mathcal{M} \mid p_{r,d} \neq \psi, q_{r,d} \neq \psi\}$. For each $d \in S_\psi^r$, let $\hat{V}_{r,d}^\psi$ denote the degenerate doubly stochastic matrix obtained by removing the $p_{r,d}$-th row and the $p_{r,d}$-th column from $V_{r,d}$. Let $\hat{\mathcal{V}} = \{\hat{V}_{r,d}^\psi \mid d \in S_\psi^r\}$, which we refer to as the restricted family. Then, for every $\psi$, its joint support graph $\mathcal{G}_\psi^r = \bigcup_{d \in S_\psi^r} \mathbf{supp}(\hat{V}_{r,d}^\psi)$ is a strongly connected graph on the vertex set $\{1, 2, \ldots, N_r\} \setminus \{\psi\}$.

Further considering the EHM state (Erdős & Simonovits, 1983; Jiang & Longbrake, 2022; Ferber et al., 2020) and the lemma of doubly stochastic matrices, we have:

$$\min_\kappa (B_\kappa)_{a,b} \geq (\hat{B}_\kappa) \geq \epsilon > 0, \tag{21}$$

where $\hat{B}$ is computed in $\mathcal{M}$. $\qquad\square$

After establishing Lemma A.1, the next step is to characterize how the Markov process avoids entering absorbing states. This requirement is equivalent to ensuring that each state transition in the Markov process is governed by a transition probability that is strictly less than 1. A more mathematically rigorous description is as follows:

Let $\{M_\psi\}_{\psi \geq 0}$ be an irreducible, aperiodic, and homogeneous Markov chain on the finite state space $\mathcal{M}$, with $P$ denoting its state transition matrix. Then there exist an integer $\overline{\kappa} \geq 1$, a sequence $\overline{H} \in \mathcal{M}^{\overline{\kappa}}$, and a constant $q \leq 0$ such that:

$$\Pr((S_{t+1}, \ldots, S_{t+\overline{\kappa}}) \in \overline{H}) \geq q. \tag{22}$$

The proof of this property is straightforward and can be obtained by only considering Lemma A.1 and the properties of Markov chains:

$$\Pr((S_{t+1}, \ldots, S_{t+\overline{\kappa}}) \in \overline{H}) = \sum_{\overline{h} \in \overline{H}} \prod_{i=1}^{\overline{\kappa}} P_{\overline{h}^{\overline{\kappa}}-1, \overline{h}^{\overline{\kappa}}} \geq 0. \tag{23}$$

After completing the analysis of the non-absorbing states of the Markov chain, we specify the corresponding non-absorption probability, which is controlled by $q$ under the assumption of non-absorbing dynamics. Given these well-constrained

conditions, the next step is to examine whether a stationary distribution exists when the exchange layer of the DecFus algorithm is governed by an external Markov process. This analysis is nontrivial: traditional convergence arguments typically rely on favorable spectral-radius properties, yet such properties are not currently available in our setting. Consequently, we must carefully estimate the spectral window, leading to the formulation of Lemma A.2:

**Lemma A.2.** *Consider an external Markov chain $\{M_\psi\}$. For a vectorized Banach space, let $z^{(\psi)} = vec(\chi^\psi) \in \Delta^n$, we have:*

$$z^\psi = V_{M_\psi} z^{\psi-1}. \tag{24}$$

*Furthermore, there must exist an iterative fixed point $z^* \in \Delta^n$ such that $V_j z^* = z^*, \forall j \in S$, and $z^* = \frac{1_n^t 1_n}{n}$.*

*Proof.* Construct the difference sequence $\Delta^\psi = z^\psi - z^*$, then we have: $\Delta^\psi = V_{M_\psi} \Delta^{\psi-1}$. Considering the construction of an upper bound for the cumulative product of $V$, we then have:

$$\min_\kappa (B_\kappa)_{i,j} \leq \min_\kappa (B_\kappa)_\kappa \sum_\kappa C_\kappa \leq \sum_\kappa (B_\kappa C_\kappa). \tag{25}$$

Here, $C_\kappa$ is an arbitrary doubly stochastic matrix. Therefore, considering the total variation distance, we have:

$$\mathbf{TV} \|C_r\|_2 \leq (1 - n\alpha) \|r\|_2, \tag{26}$$

where $\alpha$ is the minimum element in $C_r$, thus we have:

$$\mathbf{TV} \|\overline{B}_{\overline{hr}}\|_2 \leq (1 - n\epsilon) \|x\|_2. \tag{27}$$

Consider a block product process $\Phi(t + \psi : t + 1) = V_{M_{t+\psi}} \ldots V_{M_{t+1}}$. Considering applying $\Phi(t + \psi : t + 1)$ to the difference recurrence, we then have:

$$\Delta^{t+\psi} = \Phi(t + \psi : t + 1)\Delta^t,$$
$$\|\Delta^{t+\psi}\|_2 \leq (1 - n\epsilon) \|\Delta^t\|_2. \tag{28}$$

Further taking the conditional expectation of the square:

$$\begin{aligned} E[\|\Delta^{t+\psi}\|_2^2] &\leq (q(1 - n\epsilon)^2 + (1 - q)) \|\Delta^t\|_2^2 \\ &\leq \|1 - q(1 - n\epsilon)^2 \|\Delta^t\|_2^2 \\ &= \beta \|\Delta^t\|_2^2. \end{aligned} \tag{29}$$

Considering the iteration $\psi = em + M$, where $e$ is the number of cycles and $m$ is an iteration state within the Markov chain, we then have:

$$E\left[\|\Delta^t\|_2^2 \leq \beta^{\frac{\psi}{m}}\right] \|\Delta^0\|_2^2, \tag{30}$$

where $\beta = 1 - q(1 - n\epsilon)^2$ controls the iteration speed.

Let $\iota = \beta^{\frac{1}{m}}$, then we have:

$$E\|z^\psi - z^*\|_2^2 \leq \|z^0 - z^*\|_2^2 \iota^\psi. \tag{31}$$

$\square$

If we denote $\|z^0 - z^*\|_2^2 \iota^\psi = \widetilde{\rho}^\psi$, then $\widetilde{\rho}$ is the magnitude of the spectral radius during the iterative process.

A.1.3. DETAIL PROOF OF THEOREM 5.8

First, we present the expected change for each client at every iteration:

$$
\begin{aligned}
&\mathbb{E}f\left(\frac{\theta^{t+1}\mathbf{1}_N}{N}\right) \\
=&\mathbb{E}f\left(\frac{\theta^t V^t \mathbf{1}_N}{N} - \eta\frac{\nabla F(\theta^t)\mathbf{1}_N}{N}\right) \\
=&\mathbb{E}f\left(\frac{\theta^t\mathbf{1}_N}{N} - \eta\frac{\nabla F(\theta^t)\mathbf{1}_N}{N}\right) \\
\leq&\mathbb{E}f\left(\frac{\theta_t\mathbf{1}_N}{N}\right) - \eta\mathbb{E}\left\langle\nabla f\left(\frac{\theta^t\mathbf{1}_N}{N}\right), \frac{\nabla f(\theta^t)\mathbf{1}_N}{N}\right\rangle + \frac{\eta^2 L}{2}\mathbb{E}\left\|\frac{1}{N}\sum_{i=1}^{N}\nabla F_i(\theta_i^t, \xi_i^t)\right\|^2 \\
=&\mathbb{E}f\left(\frac{\theta^t\mathbf{1}_N}{N}\right) - \eta\mathbb{E}\left\langle\nabla f\left(\frac{\theta^t\mathbf{1}_N}{N}\right), \frac{\nabla f(\theta^t)\mathbf{1}_N}{N}\right\rangle \\
&+ \frac{\eta^2 L}{2}\left[\mathbb{E}\left\|\frac{\sum_{i=1}^{N}\nabla F_i(\theta_i^t) - \sum_{i=1}^{N}\nabla f_i(\theta_i^t)}{N}\right\|^2 + \mathbb{E}\left\|\frac{\sum_{i=1}^{N}\nabla f_i(\theta_i^t)}{N}\right\|^2\right] \\
=&\mathbb{E}f\left(\frac{\theta^t\mathbf{1}_N}{N}\right) - \frac{\eta - \eta^2 L}{2}\mathbb{E}\left\|\frac{\nabla f\left(\theta^t\right)\mathbf{1}_N}{N}\right\|^2 - \frac{\eta}{2}\mathbb{E}\left\|\nabla f\left(\frac{\theta^t\mathbf{1}_N}{N}\right)\right\|^2 \\
&+ \frac{\eta^2 L}{2}\underbrace{\mathbb{E}\left\|\frac{\sum_{i=1}^{N}\nabla F_i(\theta_i^t) - \sum_{i=1}^{N}\nabla f_i(\theta_i^t)}{N}\right\|^2}_{T_1} + \frac{\eta}{2}\underbrace{\mathbb{E}\left\|\nabla f\left(\frac{\theta_t\mathbf{1}_N}{N}\right) - \frac{\nabla f(\theta_t)\mathbf{1}_N}{N}\right\|^2}_{T_2}.
\end{aligned}
\tag{32}
$$

The derivations in the first and second lines rely on the asymptotic convergence property of $V$ established in Lemma 5.6. The third line follows from the $L$-smoothness assumption. Thereafter, the gradient information is decomposed into $T_1$ and $T_2$, and we first analyze the bound of $T_1$.

$$
\begin{aligned}
T_1 =&\frac{1}{N^2}\sum_{i=1}^{N}\mathbb{E}\left\|\nabla F_i(\theta_i^t) - \nabla f_i(\theta_i^t)\right\|^2 \\
&+ \frac{2}{N^2}\sum_{i=1}^{N}\sum_{i'=i+1}^{N}\mathbb{E}\langle\nabla F_i(\theta_i^t) - \nabla f_i(\theta_i^t), \nabla F_{i'}(\theta_{i'}^t) - \nabla f_{i'}(\theta_{i'}^t)\rangle \\
=&\frac{1}{N^2}\sum_{i=1}^{N}\mathbb{E}\left\|\nabla F_i(\theta_i^t) - \nabla f_i(\theta_i^t)\right\|^2 \\
&+ \frac{2}{N^2}\sum_{i=1}^{N}\sum_{i'=i+1}^{N}\mathbb{E}\langle\nabla F_i(\theta_i^t) - \nabla f_i(\theta_i^t), \mathbb{E}\left[\nabla F_{i'}(\theta_{i'}^t) - \nabla f_{i'}(\theta_{i'}^t)\right]\rangle \\
=&\frac{1}{N^2}\sum_{i=1}^{N}\mathbb{E}\left\|\nabla F_i(\theta_i^t) - \nabla f_i(\theta_i^t)\right\|^2 \\
=&\frac{1}{N}\sigma_g.
\end{aligned}
\tag{33}
$$

We now proceed to further analyze the term associated with $T_2$.

$$
\begin{aligned}
T_2 =& \mathbb{E}\left\|\nabla f\left(\frac{\theta_t \mathbf{1}_N}{N}\right) - \frac{\nabla f(\theta_t)\mathbf{1}_N}{N}\right\|^2 \\
\leq& \frac{1}{N}\sum_{i=1}^{N}\mathbb{E}\left\|\nabla f_i\left(\frac{\sum_{i'=1}^{N}\theta_{i'}^t}{N}\right) - \nabla f_i\left(\theta_i^t\right)\right\|^2 \\
\leq& \frac{L^2}{N}\sum_{i=1}^{N}\mathbb{E}\left\|\frac{\sum_{i'=1}^{N}\theta_{i'}^t}{N} - \theta_i^t\right\|^2 \\
=& \frac{L^2}{N}\sum_{i=1}^{N}\mathbb{E}\left\|\frac{\theta^t\mathbf{1}_N}{N} - \theta^t e_i\right\| \\
=& \mathbb{E}\left\|\frac{\theta^{t-1}\mathbf{1}_N - \eta\nabla F(\theta^{t-1})\mathbf{1}_N}{n} - \left(\theta^{t-1}V^{t-1}e_i - \eta\nabla F(\theta^{t-1})e_i\right)\right\|^2 \\
=& \frac{L^2}{N}\sum_{i=1}^{N}\mathbb{E}\left\|\frac{\theta^{t-1}\mathbf{1}_N - \eta\nabla F(\theta^{t-1})\mathbf{1}_N}{n} - \left(\theta^{t-1}V^{t-1}e_i - \eta\nabla F(\theta^{t-1})e_i\right)\right\|^2 \\
=& \frac{L^2}{N}\sum_{i=1}^{N}\mathbb{E}\left\|\frac{\theta^0\mathbf{1}_N - \sum_{t=0}^{T}\eta\nabla F(\theta^t)\mathbf{1}_N}{n} - \left(\theta^0 V^t e_i - \sum_{j=0}^{T-1}\eta\nabla F(\theta^j)V^{T-j-1}e_i\right)\right\|^2 \qquad (34)\\
=& \frac{L^2}{N}\sum_{i=1}^{N}\mathbb{E}\left\|\theta^0\left(\frac{\mathbf{1}_N}{n} - V^T e_i\right) - \sum_{j=0}^{T-1}\eta\nabla F(\theta^j)\left(\frac{\mathbf{1}_N}{N} - V^{T-j-1}e_i\right)\right\|^2 \\
=& \frac{\eta^2 L^2}{N}\sum_{i=1}^{N}\mathbb{E}\left\|\sum_{j=0}^{T-1}\nabla F(\theta^j)\left(\frac{\mathbf{1}_N}{N} - V^{T-j-1}e_i\right)\right\|^2 \\
\leq& 2\eta^2\,\mathbb{E}\underbrace{\left\|\sum_{j=0}^{T-1}\left(\nabla F(\theta^j) - \nabla f(\theta^j)\right)\left(\frac{\mathbf{1}_N}{N} - V^{T-j-1}e_i\right)\right\|^2}_{T_3} \\
& + 2\eta^2\,\mathbb{E}\underbrace{\left\|\sum_{j=0}^{T-1}\nabla f(\theta^j)\left(\frac{\mathbf{1}_N}{N} - V^{T-j-1}e_i\right)\right\|^2}_{T_4}.
\end{aligned}
$$

First, the consistency error is transformed into the gradient differences of each client by applying Jensen's inequality and the upper bound of component-wise summation. Then, using the $L$-Lipschitz property of the gradient, the expression is iteratively decomposed into a noise accumulation error and a propagation accumulation error.

We now proceed to further analyze the term associated with $T_3$.

$$
\begin{aligned}
T_3 &= \mathbb{E}\left\|\sum_{j=0}^{T-1}\left(\nabla F(\theta^j) - \nabla f(\theta^j)\right)\left(\frac{\mathbf{1}_N}{N} - V^{T-j-1}e_i\right)\right\|^2 \\
&= \sum_{j=0}^{T-1}\mathbb{E}\left\|\left(\nabla F(\theta^j) - \nabla f(\theta^j)\right)\left(\frac{\mathbf{1}_N}{N} - V^{T-j-1}e_i\right)\right\|^2 \\
&\leq \sum_{j=0}^{T-1}\mathbb{E}\left\|\left(\nabla F(\theta^j) - \nabla f(\theta^j)\right)\right\|_F^2\left\|\left(\frac{\mathbf{1}_N}{N} - V^{T-j-1}e_i\right)\right\|^2 \\
&\leq \frac{n\sigma^2}{1-\widetilde{\rho}}.
\end{aligned}
\tag{35}
$$

We now proceed to further analyze the term associated with $T_4$.

$$
\begin{aligned}
T_4 &= \mathbb{E}\left\|\sum_{j=0}^{T-1}\nabla f(\theta^j)\left(\frac{\mathbf{1}_N}{N} - V^{T-j-1}e_i\right)\right\|^2 \\
&= \underbrace{\sum_{j=0}^{T-1}\mathbb{E}\left\|\nabla f(\theta^j)\left(\frac{\mathbf{1}_N}{N} - V^{T-j-1}e_i\right)\right\|^2}_{T_5} \\
&\quad + \underbrace{\sum_{j\neq j'}\mathbb{E}\left\langle\nabla f(\theta^j)\left(\frac{\mathbf{1}_N}{N} - V^{T-j-1}e_i\right), \nabla f(\theta^{j'})\left(\frac{\mathbf{1}_N}{N} - V^{T-j'-1}e_i\right)\right\rangle}_{T_6}.
\end{aligned}
\tag{36}
$$

We now proceed to further analyze the term associated with $T_5$.

$$
\begin{aligned}
T_5 &= \sum_{j=0}^{T-1}\mathbb{E}\left\|\nabla f(\theta^j)\left(\frac{\mathbf{1}_N}{N} - V^{T-j-1}e_i\right)\right\|^2 \\
&\leq \sum_{j=0}^{T-1}\mathbb{E}\left\|\nabla f(\theta^j)\right\|^2\left\|\left(\frac{\mathbf{1}_N}{N} - V^{T-j-1}e_i\right)\right\|^2 \\
&\leq 6\eta^2\sum_{j=0}^{T-1}\sum_{h=1}^{N}\mathbb{E}L^2(T_3 + T_4)\left\|\left(\frac{\mathbf{1}_N}{N} - V^{T-j-1}e_i\right)\right\|^2 + 3N\sigma_l^2\frac{1}{1-\widetilde{\rho}} + \\
&\quad + 3\sum_{j=0}^{T-1}\mathbb{E}\left\|\nabla f\left(\frac{\theta^j\mathbf{1}_N}{N}\right)\mathbf{1}_N^\mathsf{T}\right\|\left\|\left(\frac{\mathbf{1}_N}{N} - V^{T-j-1}e_i\right)\right\|^2.
\end{aligned}
\tag{37}
$$

We now proceed to further analyze the term associated with $T_6$.

$$
\begin{aligned}
T_6 &= \sum_{j\neq j'}^{T-1}\mathbb{E}\left\langle\nabla f(\theta^j)\left(\frac{\mathbf{1}_N}{N} - V^{T-j-1}e_i\right), \nabla f(\theta^{j'})\left(\frac{\mathbf{1}_N}{N} - V^{T-j'-1}e_i\right)\right\rangle \\
&\leq \sum_{j\neq j'}^{T-1}\mathbb{E}\left\|\nabla f(\theta^j)\left(\frac{\mathbf{1}_N}{N} - V^{T-j-1}e_i\right)\right\|\left\|\nabla f(\theta^{j'})\left(\frac{\mathbf{1}_N}{N} - V^{T-j'-1}e_i\right)\right\| \\
&\leq \sum_{j\neq j'}^{T-1}\mathbb{E}\left\|\nabla f(\theta^j)\right\|\left\|\frac{\mathbf{1}_N}{N} - V^{T-j-1}e_i\right\|\left\|\nabla f(\theta^{j'})\right\|\left\|\frac{\mathbf{1}_N}{N} - V^{T-j'-1}e_i\right\|,
\end{aligned}
\tag{38}
$$

$$
\begin{aligned}
T_6 \leq & \sum_{j \neq j'}^{T-1} \mathbb{E} \frac{\|\nabla f(\theta^j)\|^2}{2} \left\| \frac{\mathbf{1}_N}{N} - V^{T-j-1} e_i \right\| \left\| \frac{\mathbf{1}_N}{N} - N^{T-j'-1} e_i \right\| \\
& + \sum_{j \neq j'}^{T-1} \mathbb{E} \frac{\|\|\nabla f(\theta^{j'})\|^2}{2} \left\| \frac{\mathbf{1}_N}{N} - V^{T-j-1} e_i \right\| \left\| \frac{\mathbf{1}_N}{N} - V^{T-j'-1} e_i \right\| \\
\leq & \sum_{j \neq j'}^{T-1} \mathbb{E} \left( \frac{\|\nabla f(\theta^j)\|^2}{2} + \frac{\|\nabla f(\theta^{j'})\|^2}{2} \right) \rho^{k - \frac{j+j'}{2} - 1} \\
= & \sum_{j \neq j'}^{k-1} \mathbb{E} \|\partial f(X_j)\|^2 (\widetilde{\rho})^{T - \frac{i+j'}{2} - 1} \\
\leq & 6\eta^2 \sum_{j \neq j'}^{T-1} \left( \sum_{h=1}^{n} \mathbb{E} L^2 (T_3 + T_4) + \mathbb{E} \left\| \nabla f \left( \frac{X_j \mathbf{1}_N}{N} \right) \mathbf{1}_n^\top \right\|^2 \right) (\widetilde{\rho})^{T - \frac{i+j'}{2} - 1} \\
& + \sum_{j \neq j'}^{T-1} 3N\sigma^2 (\widetilde{\rho})^{T - \frac{i+j'}{2} - 1} \\
\leq & 12\eta^2 \sum_{j \neq j'}^{T-1} \left( \sum_{h=1}^{n} \mathbb{E} L^2 (T_3 + T_4) + \mathbb{E} \left\| \nabla f \left( \frac{X_j \mathbf{1}_N}{N} \right) \mathbf{1}_n^\top \right\|^2 \right) \frac{(\sqrt{(\widetilde{\rho})})^{T-j-1}}{1 - \sqrt{(\widetilde{\rho})}} \\
& + 6N\sigma_l^2 \frac{1}{(1 - \sqrt{\widetilde{\rho}})^2}.
\end{aligned}
\tag{39}
$$

We now proceed to further analyze the term associated with $T_4$.

$$
\begin{aligned}
T_4 \leq & 6\eta^2 \sum_{j=0}^{T-1} \sum_{h=1}^{N} \mathbb{E} L^2 (T_3 + T_4) \left\| \left( \frac{\mathbf{1}_N}{N} - V^{T-j-1} e_i \right) \right\|^2 + 3N\sigma_l^2 \frac{1}{1 - \widetilde{\rho}} \\
& + 3 \sum_{j=0}^{T-1} \mathbb{E} \left\| \nabla f \left( \frac{\theta^j \mathbf{1}_N}{N} \right) \mathbf{1}_N^\top \right\| \left\| \left( \frac{\mathbf{1}_N}{N} - V^{T-j-1} e_i \right) \right\|^2 \\
& + 12\eta^2 \sum_{j \neq j'}^{T-1} \left( \sum_{h=1}^{n} \mathbb{E} L^2 (T_3 + T_4) + \mathbb{E} \left\| \nabla f \left( \frac{X_j \mathbf{1}_N}{N} \right) \mathbf{1}_n^\top \right\|^2 \right) \frac{(\sqrt{(\widetilde{\rho})})^{T-j-1}}{1 - \sqrt{(\widetilde{\rho})}} \\
& + 6N\sigma_l^2 \frac{1}{(1 - \sqrt{\widetilde{\rho}})^2} \\
\leq & 6\eta^2 \sum_{j=0}^{T-1} \sum_{h=1}^{N} \mathbb{E} L^2 (T_3 + T_4) \left\| \left( \frac{\mathbf{1}_N}{N} - V^{T-j-1} e_i \right) \right\|^2 \\
& + 3 \sum_{j=0}^{T-1} \mathbb{E} \left\| \nabla f \left( \frac{\theta^j \mathbf{1}_N}{N} \right) \mathbf{1}_N^\top \right\| \left\| \left( \frac{\mathbf{1}_N}{N} - V^{T-j-1} e_i \right) \right\|^2 \\
& + 12\eta^2 \sum_{j \neq j'}^{T-1} \left( \sum_{h=1}^{n} \mathbb{E} L^2 (T_3 + T_4) + \mathbb{E} \left\| \nabla f \left( \frac{X_j \mathbf{1}_N}{N} \right) \mathbf{1}_n^\top \right\|^2 \right) \frac{(\sqrt{(\widetilde{\rho})})^{T-j-1}}{1 - \sqrt{(\widetilde{\rho})}} \\
& + 9N\sigma_l^2 \frac{1}{(1 - \sqrt{\widetilde{\rho}})^2}.
\end{aligned}
\tag{40}
$$

Since $T_4$ is already known, the range of $T_2$ can therefore be determined. By defining the cumulative error of $T_2$ for each

client, we can subsequently express the mathematical expectation of the global cumulative client error as follows:

$$
\begin{aligned}
\mathbb{E}M^t =& \frac{\mathbb{E}\sum_{i=1}^{N} T_{2,i}}{N} \\
\leq& \frac{2\eta^2 N \sigma_g^2}{1-\widetilde{\rho}} + \frac{18\eta^2 N \sigma_l^2}{(1-\sqrt{\widetilde{\rho}})^2} \\
&+ 6\eta^2 \sum_{j=0}^{T-1} \mathbb{E} \left\| \nabla f\left(\frac{\theta^j \mathbf{1}_N}{N} \mathbf{1}_N^{\mathsf{T}}\right) \right\|^2 \left( (\widetilde{\rho})^{T-j-1} + \frac{2\sqrt{\widetilde{\rho}}^{T-j-1}}{1-\sqrt{\widetilde{\rho}}} \right) \\
&+ 6\eta^2 N L^2 \sum_{j=0}^{T-1} \mathbb{E}M_j \left( (\widetilde{\rho})^{T-j-1} + \frac{2\sqrt{\widetilde{\rho}}^{T-j-1}}{1-\sqrt{\widetilde{\rho}}} \right).
\end{aligned}
\tag{41}
$$

By tracing the error accumulation term over the past time steps, the cumulative mean can be expressed as follows:

$$
\begin{aligned}
\sum_{t=0}^{T} \mathbb{E}M^t \leq& \frac{2T\eta^2 N \sigma_g^2}{1-\widetilde{\rho}} + \frac{18T\eta^2 N \sigma_l^2}{(1-\sqrt{\widetilde{\rho}})^2} \\
&+ 6\eta^2 \sum_{t=0}^{T-1}\sum_{j=0}^{T-1} \mathbb{E} \left\| \nabla f\left(\frac{\theta^j \mathbf{1}_N}{N} \mathbf{1}_N^{\mathsf{T}}\right) \right\|^2 \left( (\widetilde{\rho})^{t-j-1} + \frac{2\sqrt{\widetilde{\rho}}^{t-j-1}}{1-\sqrt{\widetilde{\rho}}} \right) \\
&+ 6\eta^2 N L^2 \sum_{t=0}^{T-1}\sum_{j=0}^{T-1} \mathbb{E}M_j \left( (\widetilde{\rho})^{t-j-1} + \frac{2\sqrt{\widetilde{\rho}}^{t-j-1}}{1-\sqrt{\widetilde{\rho}}} \right) \\
\leq& \frac{2T\eta^2 N \sigma_g^2}{1-\widetilde{\rho}} + \frac{18T\eta^2 N \sigma_l^2}{(1-\sqrt{\widetilde{\rho}})^2} \\
&+ \frac{18\eta^2}{(1-\widetilde{\rho})^2} \sum_{j=0}^{T-1} \mathbb{E} \left\| \nabla f\left(\frac{\theta^j \mathbf{1}_N}{N} \mathbf{1}_N^{\mathsf{T}}\right) \right\|^2 \\
&+ \frac{18\eta^2 N L^2}{(1-\widetilde{\rho})^2} \sum_{j=0}^{T-1} \mathbb{E}M_j.
\end{aligned}
\tag{42}
$$

Therefore, we obtain the following result:

$$
\begin{aligned}
&\mathbb{E}f\left(\frac{\theta^{t+1}\mathbf{1}_N}{N}\right) \\
\leq& \mathbb{E}f\left(\frac{\theta^t \mathbf{1}_N}{N}\right) - \frac{\eta-\eta^2 L}{2} \mathbb{E}\left\| \frac{\nabla f(\theta^t)\mathbf{1}_N}{N} \right\|^2 - \frac{\eta}{2}\mathbb{E}\left\| \nabla f\left(\frac{\theta^t \mathbf{1}_N}{N}\right) \right\|^2 \\
&+ \frac{\eta^2 L \sigma_g^2}{2N} + \frac{\eta L^2 \mathbb{E}M^t}{2}.
\end{aligned}
\tag{43}
$$

By further simplifying the expression, we arrive at Theorem 5.8.

**Theorem A.3.** *Given that Assumptions 1–3 and 4 hold, and the step size $\eta \leq \frac{1}{2L+\sigma_g\sqrt{\frac{T}{N}}}$ such that the decentralized convergence of DecFus can be expressed as follows:*

$$
\begin{aligned}
&\frac{\mathbb{E}\left[\sum_{t=0}^{T-1}\sum_{i=1}^{N} \left\| \frac{\sum_{i'=1}^{N}\theta_{i'}^t}{N} - \theta_i^t \right\|^2 \right]}{TN} \\
\leq& \frac{18\eta^2 N}{(1-\sqrt{\widetilde{\rho}})^2 \widetilde{D_2}} \left( \frac{f(0)-f^*}{\eta T} + \frac{\eta L \sigma_l^2}{2N\widetilde{D_1}} \right) + \frac{2\eta^2 N \sigma_l^2}{(1-\widetilde{\rho})\widetilde{D_2}} \\
&+ \frac{18\eta^2 N \sigma_g^2}{(1-\sqrt{\widetilde{\rho}})^2 \widetilde{D_2}} + \frac{\eta^2 L^2 N}{\widetilde{D_1}\widetilde{D_2}} \left( \frac{\sigma_l^2}{1-\widetilde{\rho}} + \frac{9\sigma_g^2}{(1-\sqrt{\widetilde{\rho}})^2} \right),
\end{aligned}
\tag{44}
$$

where $\widetilde{D_1} = \frac{1}{2} - \frac{9\sigma_l^2 L^2 n}{(1-\sqrt{\widetilde{\rho}})^2 \widetilde{D_2}}$, $\widetilde{D_2} = 1 - \frac{18\sigma_l^2}{(1-\sqrt{\widetilde{\rho}})^2 NL^2}$, $\widetilde{\rho} = \max_t \widetilde{\rho}^t$.

## A.2. Detailed Methodology of EDecFus

As introduced in the main paper, we propose EDecFus, an extended framework building upon DecFus. Specifically, in each communication round, client $i$ uses the layer selection mechanism to send requests to its neighbors in $\mathcal{G}_i$ for designated subsets of model layers. After receiving the requested model subsets from all neighbors, client $i$ uses these subsets for subsequent aggregation. With this mechanism, no client transmits or holds the full neighbor model, which reduces communication overhead and offers a degree of privacy protection by avoiding full model exposure.

### A.2.1. SELECTIVE LAYER TRANSMISSION

**Selective Vector Generation (SVG)** At each round $t$, client $i$ generates a set of binary selection vectors, denoted as $\{\vec{s}_{i,j}^{t+1}\}_{j \in \{\mathcal{G}_i \setminus i\}}$, to be used in the next round $(t+1)$. Each vector $\vec{s}_{i,j}^{t+1} \in \mathbb{R}^{1 \times R}$ specifies the layers that client $i$ will request from its neighbor $j$ for the subsequent aggregation process.

To guarantee model convergence, we impose the constraint that, for every layer $r$, client $i$ requests this layer from at least one neighbor in $\mathcal{G}_i$. The $\vec{s}_{i,j}^{t+1}$ and the constraint is defined as:

$$\vec{s}_{i,j,r}^{t+1} = \{0,1\}, \tag{45}$$

$$\text{subject to} \sum_{j \in \{\mathcal{G}_i \setminus i\}} \vec{s}_{i,j,r}^{t+1} \geq 1, \forall r \in \{1, 2, \ldots, R\}, \tag{46}$$

where $\vec{s}_{i,j,r}^{t+1}$ denotes the $r$-th element of vector $\vec{s}_{i,j}^{t+1}$. Specifically, $\vec{s}_{i,j,r}^{t+1} = 1$ indicates that client $i$ requests the $r$-th layer from neighbor $j$, otherwise $\vec{s}_{i,j,r}^{t+1} = 0$. Note that when all elements of the selection vectors $\{\vec{s}_{i,j}^{t+1}\}_{j \in \{\mathcal{G}_i \setminus i\}}$ are set to 1, client $i$ requests the complete model from each neighbor, equivalent to the original DecFus approach.

Based on selective vector generation, we obtain a new sub-graph $\mathcal{G}_i^r$ to represent the connection of the $r^{th}$ layer in the subgraph $\mathcal{G}_i$, and we have $j \in \mathcal{G}_i^r$ if $\vec{s}_{i,j,r}^{t+1} = 1$.

**Layer Subset Packaging (LSP)** To determine the specific subset of model layers to transmit to each neighbor $j$, client $i$ uses the selection vector $\vec{s}_{j,i}^t$ received from each neighbor $j$ in the previous round $t-1$ after local training. Specifically, client $i$ generates the sparse model $\hat{\theta}_{i,j}^{t+\frac{1}{2}}$ for each neighbor $j \in \{\mathcal{G}_i \setminus i\}$ based on its local pre-aggregation model $\theta_i^{t+\frac{1}{2}} = [L_{i,1}^{t+\frac{1}{2}}, L_{i,2}^{t+\frac{1}{2}}, ..., L_{i,R}^{t+\frac{1}{2}}]$, represented by:

$$\hat{\theta}_{i,j}^{t+\frac{1}{2}} = \{L_{i,r}^{t+\frac{1}{2}} | \vec{s}_{j,i,r}^t = 1, r \in [1, ...R]\}, j \in \{\mathcal{G}_i \setminus i\}. \tag{47}$$

Client $i$ then sends both the sparse model $\hat{\theta}_{i,j}^{t+\frac{1}{2}}$ and the next round $t+1$ selection vector $s_{i,j}^{t+1}$ to each neighbor $j$ without additional communication rounds.

### A.2.2. ADAPTED EDECFUS UPDATE

After receiving the subset models $\hat{\theta}_{j,i}^{t+\frac{1}{2}}$ from all neighbors $j \in \mathcal{G}_i$, client $i$ performs standard DecFus procedure with slight modification.

Based on the sub-graph $\mathcal{G}_i^r$, the **Similarity-based Layer Grouping (SLG)** is modified as

$$\sigma_{i,r} = \min_j CS(L_{i,r}^{t+\frac{1}{2}}, \tilde{L}_{j,i,r}^{t+\frac{1}{2}}), \mathbf{j} \in \{\mathcal{G_i^r} \setminus \mathbf{i}\}, \tag{48}$$

where the only modification is that the constraint is updated from $j \in \{\mathcal{G}_i \setminus i\}$ in DecFus (Equation (7) in the mian paper) to $j \in \{\mathcal{G}_i^r \setminus i\}$ in EDecFus.

Similarly, the modification of **Unified Layer-wise Aggregation** in EDecFus lies in the updated constraint $\mathcal{G}_i^r$, compared to DecFus, which is defined as:

$$L_{i,r}^{t+1} = \sum_{\mathbf{j} \in \mathcal{G_i^r}} \lambda_{ij}^r L_{j,r}^{t+\frac{1}{2}}, \tag{49}$$

$$\lambda_{ij}^r = \begin{cases} w_{ij}, & \text{if } r \in \mathcal{P}_i^t \\ 0 \text{ or } 1, & \text{if } r \in \mathcal{Q}_i^t \end{cases}, \mathbf{j} \in \mathcal{G}_i^{\mathbf{r}}. \tag{50}$$

Notably, the only difference between Equation (50) in the appendix and Equation (12) in the main paper is the inclusion of the additional constraint $\mathbf{j} \in \mathcal{G}_i^{\mathbf{r}}$.

It is worth noting that, given a sufficiently large number of global rounds, this method guarantee that the contributions of all clients are adequately considered. Specifically, the communication complexity per round is reduced at most from $\mathcal{O}(\mathbb{E}[|\mathcal{G}_i \setminus \{i\}|] \cdot R)$ — as in conventional DFL methods — to $\mathcal{O}(R)$, where $\mathbb{E}[|\mathcal{G}_i \setminus \{i\}|]$ denotes the mean number of neighbors of client $i$, and $R$ is the number of layers. Moreover, no party ever holds the full model of another, making model-based privacy attacks significantly harder.

Based on the above discuss, we proposed EDecFus as detailed in Algorithm 2.

---

**Algorithm 2** EDecFus

---

**Input:** Client set $\mathcal{N}$, communication topology $\mathcal{G}$, datasets $\{\mathcal{X}_i\}_{i \in \mathcal{N}}$, total training rounds $T$, stability threshold $\epsilon_s$
**Output:** Optimal models $\theta_i^T$ for client $i \in \mathcal{N}$
**Initialization:** Local model $\theta_i^0 = [L_{i,1}^0, ..., L_{i,r}^0, ..., L_{i,R}^0]$ for client $i \in \mathcal{N}$
**for** round $t = 1$ **to** $T$ **do**
  **for** each client $i \in \mathcal{N}$ **in parallel do**
    $\theta_i^{t+\frac{1}{2}} = [L_{i,1}^{t+\frac{1}{2}}, \ldots, L_{i,R}^{t+\frac{1}{2}}] \leftarrow$ Local training on $\mathcal{X}_i$ by (1)
    $\mathcal{G}_i^r \leftarrow \text{SVG}(\{\vec{s}_{i,j}^{t+1}\}_{j \in \mathcal{G}_i \setminus \{i\}})$ by (46)
  **end for**
  **for** each client $i \in \mathcal{N}$ **in parallel do**
    $\tau_i^t \leftarrow \text{DCD}(\epsilon_s, \mathcal{L}_j^{t-1}, j \in \{\mathcal{G}_i^r\})$ by (4),(5)
    **for** $r = 1$ **to** $R$ **do**
      $CS(L_{i,r}, L_{j,r}), j \in \{\mathcal{G}_i^r \setminus i\}$ by (6)
      $\sigma_{i,r} \leftarrow \min_j CS(L_{i,r}, L_{j,r}), j \in \{\mathcal{G}_i^r \setminus i\}$
    **end for**
    $\mathcal{P}_i^t, \mathcal{Q}_i^t \leftarrow \text{Group}\{1, 2, ..., R\}$ by $\tau_i^t, \{\sigma_{i,r}\}_{r=1}^R$
    **for** $r = 1$ **to** $R$ **do**
      **if** $r \in \mathcal{P}_i^t$ **then**
        $L_{i,r}^{t+1} \leftarrow \text{LA}(L_{j,r}^{t+\frac{1}{2}}, j \in \{\mathcal{G}_i^r\})$ by (8)
      **end if**
      **if** $r \in \mathcal{Q}_i^t$ **then**
        $L_{i,r}^{t+1} \leftarrow \text{LE}(L_{j,r}^{t+\frac{1}{2}}, j \in \{\mathcal{G}_i^r\})$ by (9)
      **end if**
    **end for**
  **end for**
**end for**

---

## A.3. Experimental Setup

To evaluate the effectiveness of DecFus and EDecFus, we conduct experiments under the decentralized communication topology illustrated in Figure 9 (a). The algorithms are evaluated on different datasets, various non-IID degree, and different model architectures. All experiments results were resulted from an Ubuntu workstation equipped with an NVIDIA RTX 3090 GPU and 32 GB of memory. Note that in both the main text and appendix, the version of EDecFus used for experiments is the one with minimum communication cost, corresponding to the case in (S2) where $\sum_{j \in \{\mathcal{G}_i \setminus i\}} \vec{s}_{i,j,r}^{t+1} = 1$. The detailed experimental settings and parameters are as follows.

### A.3.1. DATASETS

We evaluate our proposed DecFus and EDecFus on the following three benchmark datasets. As shown in Figure 8, we adopt the Dirichlet distribution $Dir(\alpha)$ to control the degree of data heterogeneity across clients, where the concentration parameter $\alpha$ determines the level of heterogeneity (Hsu et al., 2019). A larger $\alpha$ corresponds to lower heterogeneity,

indicating that the data distribution is closer to an IID setting. We consider three Dirichlet configurations with $\alpha$ values of 1, 0.5, and 0.1, representing mild, moderate, and extreme data heterogeneity scenarios, respectively.

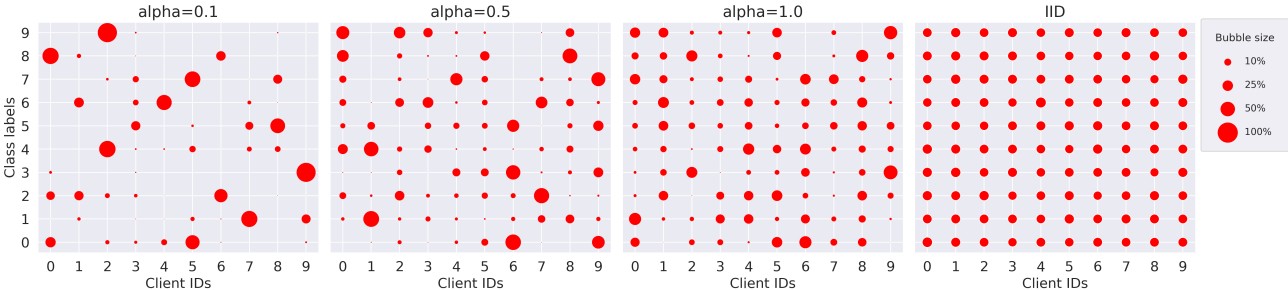

*Figure 8.* The non-IID data distribution of CIFAR-10 dataset.

- **SVHN:** A widely used dataset for digit recognition in natural scene images, SVHN contains 73,257 training images and 26,032 testing images of house numbers collected from street views. It's widely used to assess basic image classification models (Netzer et al., 2011).

- **CIFAR-10:** This dataset consists of 60,000 images spanning 10 distinct classes, including various animals and vehicles. Each class contains 5,000 training images and 1,000 test images. CIFAR-10 serves as a standard benchmark for evaluating models on slightly more challenging classification task (Krizhevsky & Hinton, 2009).

- **CIFAR-100:** CIFAR-100 is an image classification dataset containing 60,000 color images grouped into 20 coarse categories. Each category has 3,000 images, with 2,500 for training and 500 for testing. It serves as a benchmark for evaluating models on more diverse and challenging classification tasks (Krizhevsky & Hinton, 2009).

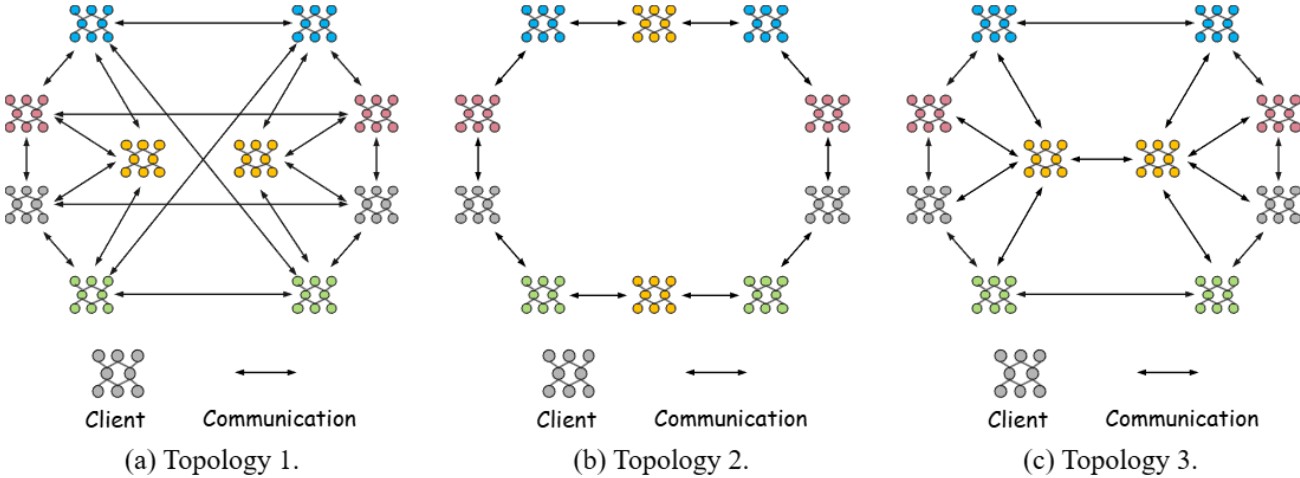

*Figure 9.* Evaluated communication topologies.

### A.3.2. HYPER-PARAMETERS

For fair comparison, all evaluated methods set the SGD optimizer with a learning rate of 0.01 and a momentum of 0.5. Each client performs 3 epochs of local training in every communication round. The batch size is set to 64 for CIFAR-10/100 and 128 for SVHN, with the total number of global rounds set to 1000 and 500, respectively. The other hyperparameters used in our experiments include the early training rounds $\mathcal{T}_r = 200$ for CIFAR-10/100 and 100 for SVHN, a upper cutoff threshold of $\tau_U = 0.75$ and a lower threshold of $\tau_L = 0.25$.

### A.3.3. ROBUSTNESS ON DIFFERENT DECENTRALIZED COMMUNICATION TOPOLOGIES

To comprehensively assess the influence of communication topology, we conduct evaluations under different topologies and use the same method from the main paper as baselines. We employ Topologies 1-3 for the SVHN, CIFAR-100, and CIFAR-10 datasets, respectively. The figure 9 shows Topologies 1-3.

*Table 6.* Average accuracy $\pm$ standard deviation of all clients across SVHN, CIFAR-100, and CIFAR-10 datasets, evaluated under different communication topologies.

| Topology | Dataset | Heter. Set. | Test Accuracy (%) | | | | | | | |
|---|---|---|---|---|---|---|---|---|---|---|
| | | | DFLAvg | DFLProx | DFLExP | DisPFL | DFedSAM | DFLMR | DecFus | EDecFus |
| Topology 1 | SVHN | 0.1 | 53.32±0.59 | 54.10±0.78 | 51.60±0.97 | 52.24±1.73 | 50.51±0.51 | 54.27±0.61 | **57.83±0.85** | 56.35±0.79 |
| | | IID | 91.38±0.13 | 91.48±0.16 | 91.20±0.08 | 92.10±0.10 | 91.98±0.12 | 92.09±0.03 | **93.43±0.04** | 92.17±0.04 |
| Topology 2 | CIFAR100 | 0.1 | 41.62±2.20 | 42.05±3.01 | 43.46±1.98 | 31.02±0.69 | 42.62±3.57 | 35.01±1.57 | **43.67±3.80** | 42.45±1.90 |
| | | IID | 60.65±1.04 | 60.28±0.87 | 61.13±0.70 | 55.59±1.28 | 61.20±0.28 | 63.51±0.94 | **66.89±0.94** | 64.89±0.40 |
| Topology 3 | CIFAR10 | 0.1 | 31.56±0.82 | 31.79±0.59 | 29.24±0.50 | 31.89±0.34 | 28.56±1.35 | 31.02±0.99 | **33.81±0.96** | 33.76±0.62 |
| | | IID | 75.15±0.82 | 73.83±0.50 | 73.52±0.77 | 76.68±2.31 | 72.05±0.53 | 75.98±0.72 | **76.72±0.62** | 76.60±0.49 |

*Table 7.* Communication Overhead evaluated under different communication topologies.

| Model | Topology | Communication Overhead(MB) | | | | | | | |
|---|---|---|---|---|---|---|---|---|---|
| | | DFLAvg | DFLProx | DFLExP | DisPFL | DFedSAM | DFLMR | DecFus | EDecFus |
| ResNet-50 | Topology 1 | 900 | 900 | 900 | 1800 | 900 | 900 | 900 | **225** |
| | Topology 2 | 720 | 720 | 720 | 1440 | 720 | 720 | 720 | **372** |
| | Topology 3 | 450 | 450 | 450 | 900 | 450 | 450 | 450 | **225** |
| VGG16 | Topology 1 | 5134 | 5134 | 5134 | 10268 | 5134 | 5134 | 5134 | **1284** |
| | Topology 2 | 4107 | 4107 | 4107 | 8214 | 4107 | 4107 | 4107 | **2132** |
| | Topology 3 | 2567 | 2567 | 2567 | 5134 | 2567 | 2567 | 2567 | **1284** |

Table 6 presents the quantitative results of our DecFus and EDecFus alongside the state-of-the-art baselines over topologies 1-3 on three datasets. It is seen from the table that our proposed DecFus still outperforms all baselines across different communication topologies with various non-IID degrees $\alpha$. This further verifies the robustness of the proposed DecFus on different decentralized communication topologies.

### A.3.4. COMMUNICATION AND PRIVACY COMPARISON

Table 7 presents a comparison of communication overhead between our proposed EDecFus and several baseline methods under different communication topologies in CIFAR-10. As mentioned previously, the communication cost of EDecFus is related to the average number of neighbors per client and consistently demonstrates reduced communication overhead across all communication topologies. Note that the high communication cost of DisPFL is attributed to its additional transmission of sparse masks alongside model parameters. Additionally, this selective transmission strategy provides resistance to Deep Leakage from Gradients (DLG) (Zhu et al., 2019) attacks as described in (Ding et al., 2024).

### A.3.5. COMPUTATION OVERHEAD (FLOPs) COMPARISON

*Table 8.* Computation overhead (FLOPs) evaluated under different algorithms.

| Model | Computational Cost | Floating Point Operations (TFLOPs) | | | | | | | |
|---|---|---|---|---|---|---|---|---|---|
| | | DFLAvg | DFLProx | DFLExP | DisPFL | DFedSAM | DFLMR | DecFus | EDecFus |
| ResNet50 | Baseline Training Computation | 11.11 | 11.11 | 11.11 | 11.11 | 11.11 | 11.11 | 11.11 | 11.11 |
| | Algorithm-induced Overhead | 0 | 0 | 1.26e-5 | 1.88e-5 | 11.11 | 0 | 7.51e-5 | 7.51e-5 |
| VGG16 | Baseline Training Computation | 19.96 | 19.96 | 19.96 | 19.96 | 19.96 | 19.96 | 19.96 | 19.96 |
| | Algorithm-induced Overhead | 0 | 0 | 13.45e-5 | 20.17e-5 | 19.96 | 0 | 80.70e-5 | 80.70e-5 |

To further assess the computational efficiency of different algorithms, we report the computation overhead measured by floating-point operations (FLOPs). Specifically, we decompose the per-round cost into the baseline training computation per client, which include standard forward and backward passes for local training, and the algorithm-induced overhead introduced by each strategy which include additional similarity computation, sparse masking, or perturbation-related operations. The results are summarized in Table 8.

Overall, the baseline local training computation dominates the total FLOPs across most methods, while the algorithm-induced overhead remains negligible in comparison. In particular, the additional computational load introduced by DecFus and EDecFus is several orders of magnitude smaller than the baseline training cost, confirming that the proposed similarity mechanisms incur negligible computational overhead. Besides, we also observe that DFedSAM involving extra perturbation-

related steps introduce noticeably higher overhead, consistent with the need for additional computation beyond a single standard training pass.

