# OpenReview forum: "DecFus: Decentralized Layer-wise Fusion with Dynamic Exploration and Exploitation"
_ICML.cc/2026/Conference — ICML 2026 regular_

### Official Review · Reviewer_bSov · 2026-03-05

**Soundness:** 3
**Presentation:** 3
**Significance:** 3
**Originality:** 3
**Overall Recommendation:** 5
**Confidence:** 3

**Summary:**

This paper focuses on Decentralized Layer-wise Fusion (DecFus) which proposes the first DFL framework that unifies layer-level exchange and averaging to balance exploration and exploitation. In specifics, to address the exploration deficiency of conventional model averaging methods, authors of this paper develops a framework called "Decentralized Layer-wise Fusion (DecFus)" which consists of two parts: 1)  layer exchange that serves as exploration for exploring larger region and escaping from sharp ravine; 2) layer averaging that constitutes exploitation for steering the model toward more stable and potentially better solutions. To further boost performances, authors devise a dynamic threshold to balance the ratio between layer exchange and layer averaging. Besides these empirical designs, authors further give several theoretical results for validating the convergence properties of the algorithm.

**Compliance With Llm Reviewing Policy:**

Affirmed.

**Final Justification:**

The rebuttal addressed main concerns.

The authors added more ablations in NLP tasks for validating effectiveness of proposed algorithm, thereby making the whole story very sound.

**Key Questions For Authors:**

Did you try other architectures besides ResNet/VGG? And did you try tasks besides image classification tasks like NLP?

I hope the authors can use more experiments (if possible) to show the generality of proposed framework.

**Limitations:**

Yes.

**Strengths And Weaknesses:**

Strengths:
1. The motivation is well explained and the presentation for designs of algorithm is easy to be understood.

2. Besides the design and experiments, authors also make several theorems to analyze the convergence of the proposed algorithms, thereby making the whole paper very sound.

3. Compared to baselines, authors propose layer exchanges to further augment exploration for exploring larger region. This point is novel.

Weaknesses:
1. I think layer merging is very similar to the model merging paradigm. Could you compare your framework with model merging in the literature. I think they have a lot of common points.

2. The experiments adopt VGG and ResNet as backbones to validate the effectiveness of proposed algorithm. To show its generality, I think authors should also try some other backbones like Bert/CLIP if possible.

---

> ### Author Rebuttal · Authors · 2026-03-30
>
> We thank you for the positive and constructive comments. We are encouraged for your recognition of this work.
>
> **Q1:** I think layer merging is very similar to the model merging paradigm. Could you compare your framework with model merging in the literature. I think they have a lot of common points.
>
> **A1:** You are correct that layer merging and model merging share similarities, as both involve combining parameters from different clients.
>
> However, the key difference lies in the granularity of aggregation. Model merging is typically performed at the whole-model level, as in conventional FL methods such as FedAvg and its variants (e.g., FedProx), where complete client models are aggregated into a single shared model~\cite{[1,2]}.
>
> In contrast, DecFus operates at the layer level and does not uniformly merge entire models. Instead, it differentiates across layers. That is, some layers are exchanged to preserve diversity and encourage exploration, while some other layers are averaged to promote stability and exploitation. This layer-wise design enables the aggregation behavior to adapt to heterogeneous alignment across layers and clients.
>
> Therefore, although DecFus is related to model merging in the general sense of merging parameters, it departs from standard model-level aggregation by introducing a dynamic, layer-wise fusion mechanism. We will clarify this distinction in the revision.
>
>
>
> **Q2:** The experiments adopt VGG and ResNet as backbones to validate the effectiveness of proposed algorithm. To show its generality, I think authors should also try some other backbones like Bert/CLIP if possible.
>
> **A2:** We thank you for suggesting a more detailed evaluation on a broader range of tasks would provide stronger evidence of DecFus generality.
>
> To further verify the generality of DecFus, we conducted an additional experiment using MobileBERT on AG News, which differs substantially from the image classification tasks in the current draft.
>
> ||DecFus|DFLAvg|DFLMR|
> |--|--|--|--|
> |IID|93.83±0.16|93.73±0.17|93.52±0.19
> |non-IID $\alpha=0.1$|50.06±0.84|49.06±0.98|48.56±1.16
>
> The result shows that DecFus still outperforms the compared methods in this text classification setting, suggesting that the advantage of DecFus extends across different task objectives rather than being specific to the vision classification setting studied in the current submission. Due to the limited rebuttal time, we were able to include only a subset of representative baselines in this additional experiment, but the result still provides supportive evidence for the broader applicability of the proposed framework.
>
> In the revision, we will include this additional result and clarify more explicitly that DecFus is applicable beyond the image classification setting currently reported in the paper.
>
> [1] B McMahan, E Moore, D Ramage, S Hampson, BA y Arcas, Communication-Efficient Learning of Deep Networks from Decentralized Data. PMLR 2017.
> [2] T Li, AK Sahu, M Zaheer, M Sanjabi, A Talwalkar, V Smith, Federated Optimization in Heterogeneous Networks. MLSys 2020.

---

> > ### Author Rebuttal · Reviewer_bSov · 2026-04-01
> >
> > My concerns have been adequately addressed. I keep rated score.

---

> > > ### Author Response · Authors · 2026-04-07
> > >
> > > Thanks for your insightful comments and for taking the time to review our rebuttal. We are pleased that our response has adequately addressed your concerns, and we will incorporate these clarifications and additional results in our final version.
> > >
> > > If you feel that these additions improve the overall quality of the work, we would be grateful if you might consider this in your final score. In any case, we sincerely appreciate your time and valuable feedback.

---

### Official Review · Reviewer_yDmK · 2026-03-11

**Soundness:** 3
**Presentation:** 2
**Significance:** 3
**Originality:** 3
**Overall Recommendation:** 4
**Confidence:** 4

**Summary:**

This  paper  studies decentralized learning, where a global model is trained collaboratively without relying on a central server. A key challenge in this architecture is ensuring efficient propagation of model information across the communication graph. Without sufficient mixing between clients, learning may become restricted to local regions of the graph, leading each client to train its model using information from only a limited subset of participants. This work modifies the way each client generates a new model from its own parameters and those of its neighbors. Instead of relying  on the traditional averaging of models, the proposed method improves exploration by introducing layer swapping. However, this strategy may also lead to instability due to highly diverse update directions.
The main idea behind the proposed method, called  DecFus,  is therefore to define two aggregation modes: swapping and averaging. Depending on the divergence of each layer with respect to the corresponding layers of its neighbors, the aggregation strategy is adapted. When the layers are well aligned, the method favors averaging, which corresponds to the standard aggregation mechanism. Conversely, when the layers are poorly aligned, the method replaces the layer with one taken from a neighbor through a swapping operation.

The objective is to enhance each client's ability to explore the communication graph and thus improve the chances of converging to the optimal solution. Since the learning dynamics vary across layers during training, the method introduces a learning process composed of two phases: exploration and exploitation. These phases are controlled using the variance of the losses observed among neighboring clients. Furthermore, the convergence of DecFus is theoretically established without requiring the aggregation matrix to be doubly stochastic. Extensive experiments show that DecFus outperforms existing CFL and DFL schemes across multiple datasets and experimental settings.

**Compliance With Llm Reviewing Policy:**

Affirmed.

**Final Justification:**

I appreciate the authors’ efforts in evaluating the performance of their approach against the topology I proposed. The results clearly indicate that the achieved accuracy is lower than what the authors reported in their prior work on other topologies.

Overall, I find the contribution of this article to be valuable. However, I recommend that the authors further strengthen the manuscript by explicitly discussing the limitations of their method, particularly when applied to topologies that were not considered in the initial version of the work.

**Key Questions For Authors:**

See the weak points that need to be addressed in this article. I also have a few additional points that need to be clarified in the paper:


It would be useful to clarify how the exploration–exploitation transition is controlled over time and how sensitive the method is to the chosen thresholds.
Some metrics should defined mathematically to understand the meaning, for example  “aggregation shift”
Certain layers may be inherently harder to train, leading to misalignment across clients. In the proposed aggregation strategy, such layers may be swapped more often than averaged, potentially increasing instability and communication overhead. The paper should clarify how it mitigates this issue and whether the method includes safeguards to handle layers that consistently diverge.

**Limitations:**

yes

**Strengths And Weaknesses:**

Strengths:

- The paper proposes an original approach to decentralized federated learning (DFL) by introducing a mechanism that balances exploration and exploitation across the communication graph.

- The adaptive strategy that alternates between layer averaging and layer swapping depending on the alignment between client layers is interesting and well motivated.

- Exploration-Exploitation control: The dynamic control of exploration during training, based on the variance of the losses among neighbors, is a promising mechanism to improve model propagation in the graph.

- The convergence analysis appears solid and rigorous, especially since it does not require the aggregation matrix to be doubly stochastic, which is a strong assumption in many decentralized learning methods.

Weaknesses

- This work does not clearly specify for which types of network topology the proposed solution is expected to perform well or poorly. For example, consider a graph composed of two clusters connected through a small number of hub nodes. Since each node is primarily influenced by the models within its own cluster, it is unclear whether the proposed method would effectively enable exploration of other clusters in the graph. In such scenarios, information propagation across clusters may remain limited. Therefore, the nature of the communication topology plays an important role, and the paper does not clearly discuss whether the proposed approach is suitable for arbitrary graph structures.

- In the experimentation phase, it is important to verify whether all clients converge to the same model, i.e., whether consensus is achieved. Reporting accuracy alone is not sufficient, as it does not reveal whether the clients’ models have aligned. Ideally, experiments should include explicit measures or visualizations to indicate whether consensus has been reached across the network

- The proposed solution introduces additional computational overhead, yet this extra cost is not quantified or analyzed in the paper. Understanding the magnitude of this overhead is important to evaluate the practical feasibility and efficiency of the method.

- Certain layers may be inherently harder to train, leading to misalignment across clients. In the proposed aggregation strategy, such layers may be swapped more often than averaged, potentially increasing instability and communication overhead. The paper should clarify how it mitigates this issue and whether the method includes safeguards to handle layers that consistently diverge.

---

> ### Author Rebuttal · Authors · 2026-03-31
>
> We appreciate your positive feedback and insightful comments.
>
> **Q1:** Topology sensitivity discussion.
>
> **A1:** For you suggest sparse inter-cluster connections, we added additional experiments, under(SVHN, ResNet50, non-IID with $\alpha=1$), with adjacency matrices respectively as
> $$
> A^{(4)} =
> \begin{bmatrix}
> 0 & 1 & 1 & 0 & 0 & 0 & 0 & 0 & 0 & 0\\\\
> 1 & 0 & 1 & 1 & 0 & 0 & 0 & 0 & 0 & 0\\\\
> 1 & 1 & 0 & 1 & 1 & 0 & 0 & 1 & 0 & 0\\\\
> 0 & 1 & 1 & 0 & 1 & 0 & 0 & 0 & 0 & 0\\\\
> 0 & 0 & 1 & 1 & 0 & 0 & 0 & 0 & 0 & 0\\\\
> 0 & 0 & 0 & 0 & 0 & 0 & 1 & 1 & 0 & 0\\\\
> 0 & 0 & 0 & 0 & 0 & 1 & 0 & 1 & 1 & 0\\\\
> 0 & 0 & 1 & 0 & 0 & 1 & 1 & 0 & 1 & 1\\\\
> 0 & 0 & 0 & 0 & 0 & 0 & 1 & 1 & 0 & 1\\\\
> 0 & 0 & 0 & 0 & 0 & 0 & 0 & 1 & 1 & 0
> \end{bmatrix},
> $$
>
> $$
> A^{(5)} =
> \begin{bmatrix}
> 0 & 1 & 1 & 0 & 0 & 0 & 0 & 0 & 0 & 1\\\\
> 1 & 0 & 1 & 1 & 0 & 0 & 0 & 0 & 0 & 0\\\\
> 1 & 1 & 0 & 1 & 1 & 0 & 0 & 1 & 0 & 0\\\\
> 0 & 1 & 1 & 0 & 1 & 0 & 0 & 0 & 0 & 0\\\\
> 0 & 0 & 1 & 1 & 0 & 1 & 0 & 0 & 0 & 0\\\\
> 0 & 0 & 0 & 0 & 1 & 0 & 1 & 1 & 0 & 0\\\\
> 0 & 0 & 0 & 0 & 0 & 1 & 0 & 1 & 1 & 0\\\\
> 0 & 0 & 1 & 0 & 0 & 1 & 1 & 0 & 1 & 1\\\\
> 0 & 0 & 0 & 0 & 0 & 0 & 1 & 1 & 0 & 1\\\\
> 1 & 0 & 0 & 0 & 0 & 0 & 0 & 1 & 1 & 0
> \end{bmatrix},
> $$
> The results are as follows, which further demonstrates the superior performance of DecFus.
> ||DecFus|DFLAvg|DFLMR
> |-|-|-|-|
> |Topology 4|86.91±1.01|83.64±2.57|84.89±2.01
> |Topology 5|91.02±0.58|83.37±1.59|85.12±1.07
>
> To further examine the impact of topology on DecFus beyond the robustness results in the appendix, we analyze the relationship between algebraic connectivity and model performance. We observe that performance varies with algebraic connectivity across topologies, broadly consistent with prior DFL literature~\cite{[1]}.
>
>
> |Topology |1|2|3|4|5
> |-|-|-|-|-|-|
> |Algebraic connectivity|1.875|0.382|0.895|0.298|0.551
> |Accuracy|89.31±0.68|89.60±1.04|90.37±2.75|86.91±1.01|91.02±0.58
>
>
> **Q2:** Evaluation of consensus.
>
> **A2:** To verify model consensus, we additionally measured the inter-client parameter and training loss variance, as well as the coefficients of variation of accuracy, in the setting(SVHN, ResNet-50, non-IID with $\alpha=1$):
> |Round|50|150|250|350|450
> |-|-|-|-|-|-|
> |Model Parameters Variance|0.0174|0.0022|0.0012|0.0009|0.0007
> |Test accuracy Coefficients Variation|0.182|0.120|0.091|0.076|0.068
> |Training loss Variance|0.36|0.14|0.11|0.07|0.04
>
> In all cases, the variance decreases over training, indicating that client models and predictions become progressively more aligned, which demonstrates the consensus of DecFus.
>
> **Q3:** Computational overhead analyze.
>
> **A3:**
> The additional overhead of DecFus mainly arises from similarity computation, with per-round complexity
> $\mathcal{O}\left( C_{|G_i|}^{2} \cdot \sum_{r=1}^{R} d_r \right)$, compared to $\mathcal{O}( E \cdot \frac{\vert X_i \vert}{B} \cdot \sum_{r=1}^{R} d_r )$ for local training. The corresponding TFLOPS are as follows
>
> |Computational Cost| ResNet50 | VGG16 |
> |-|-|-|
> |Training Computation|11.1168 TFLOPs|19.961 TFLOPs
> |Algorithmic-induced Overhead|7.5158e-5 TFLOPs|80.7019e-5 TFLOPs
>
> Since similarity computation is lightweight and independent of epochs and batches, its additional cost is negligible. Consequently, the overhead introduced by DecFus is orders of magnitude smaller than that of local training.
>
>
> **Q4:** Control for consistently divergent layers.
>
> **A4:** DecFus mitigates this issue by gradually reducing layer exchange and increasing layer averaging, so that persistently difficult layers are less likely to remain exchange-dominated in later stages. We also counted how often each layer was averaged during training, and found that every layer was averaged at least one times, indicating that no layer stays in a persistent exchange-only pattern.
>
> **Q5:** Transition control and threshold sensitivity.
>
> **A5:** In DecFus, the transition is governed by the dynamic cutoff $\tau_i^t$, which remains high before neighborhood stabilization to encourage exploration via exchange, and decreases according to Eq.~(5) once the stability score falls below the threshold $\epsilon_s$. This progressively shifts more layers from exchange to averaging, leading to a transition from exploration-dominant to exploitation-dominant aggregation.
> Regarding parameter sensitivity, we conducted additional experiments (see response to Reviewer 9kMC due to space limits). The results show that moderate variations in threshold-related parameters result in less than $2\%$ accuracy fluctuation.
>
>
> **Q6:** Metrics mathematically define.
>
> **A6:** To make this metric more precise, we define the aggregation shift as
>
> $S^{t} = \frac{1}{N}\sum_{i=1}^{N}\left\|\theta_i^{t+1}-\theta_i^{t+\frac{1}{2}}\right\|_2.$
>
> This quantity measures the average distance of client model parameters before and after aggregation at round $t$. We will include the define in the revision.
>
> [1] L Kong, T Lin, A Koloskova, M Jaggi, S Stich, Consensus Control for Decentralized Deep Learning. ICML 2021.

---

> > ### Author Rebuttal · Reviewer_yDmK · 2026-04-01
> >
> > My comments have been partially addressed, but several points still need further consideration:
> >
> > 1- The topology considered in this paper is rather simplistic and does not allow us to assess whether the proposed method remains effective under more complex network structures. I recommend evaluating the approach on a simple  topology consisting of  two clusters: Cluster 1 centered around Hub 1 and Cluster 2 centered around Hub 2. Each hub connects to approximately 40 nodes, forming a star topology locally.  I need to see  the performance using the proposed  approach with alpha=0.1 and Resnet 8 with CIFAR 10
> >
> > 2- For Question 2, I would like to see an analysis of the distance between the model parameters (W) across different clients, particularly for hub1–hub2 and hub1–node  or hub2-node.
> >
> > 3-A sensitivity analysis of the parameters is still necessary, and I would like the authors to conduct this analysis using the proposed topology.

---

> > > ### Author Response · Authors · 2026-04-07
> > >
> > > We thank the reviewer for the follow-up comments and apologize for the delayed response, as the additional experiments are substantially larger (approximately eight times the scale of the previous experiments).
> > >
> > >
> > > **Q1:** Performence on two-cluster topology.
> > >
> > > **A1:** It is true that the topologies considered in the paper are relatively simple, although we have included robustness experiments across various topologies.
> > > To further verify the effectiveness of DecFus, we evaluate it on your suggested two-cluster topology with 80 nodes, under the setting with $\alpha=0.1$ on CIFAR-10.
> > >
> > > Note that we adopt ResNet-18 in these experiments, as ResNet-8 is not a commonly used architecture (and may be a typo in your comment). The results are as follows.
> > >
> > > ||DecFus|DFLAvg|DFLMR
> > > |-|-|-|-|
> > > |Two-cluster Topology|27.86±0.30|24.08±0.99|24.86±0.73
> > >
> > > From the above table, we observe that DecFus consistently achieves the highest accuracy among all baselines, while DFLAvg and DFLMR exhibit similar performance. This further verifies the effectiveness of DecFus on more complex topologies.
> > >
> > >
> > >
> > > **Q2:** Model parameter distance analysis.
> > >
> > > **A2:** Following your advice, we measure the $\ell_2$ distance between model parameters across clients under the two-cluster topology. Specifically, we report the distances for Hub1–Hub2, Hub1–Node, Hub2–Node, and Avg–Node.
> > >
> > > Here, Hub1–Node denotes the mean $\ell_2$ distance between Hub1 and all non-hub clients in Cluster 1, while Hub2–Node is defined analogously for Hub2 and all non-hub clients in Cluster 2. Avg–Node is defined as the average distance between each client model and their global average model.
> > > The corresponding results are as follows.
> > >
> > >
> > > |Round|100|200|300|400|500|600|700|800|900|1000
> > > |-|-|-|-|-|-|-|-|-|-|-|
> > > |Hub1–Hub2|23.24|26.15|27.72|31.49|19.65|19.77|17.74|13.15|18.85|10.84
> > > |Hub1–Node|21.00|25.54|30.31|24.31|18.38|17.74|16.53|17.40|16.75|15.57
> > > |Hub2–Node|20.74|23.77|25.06|27.02|25.68|21.74|14.85|15.65|14.52|14.36
> > > |Avg-Node|13.37|16.49|19.62|18.37|15.78|12.12|12.23|12.27|12.95|11.71
> > >
> > > From the above table, we observe that all the above four metrics first increase and then decrease over training rounds. This trend is consistent with the design intuition of DecFus: in the early stage, layer exchange dominates for exploration, leading to larger divergence, while in later stages, layer averaging dominates for exploitation, resulting in reduced divergence.
> > >
> > > Notably, the decrease in both inter-cluster and intra-cluster distances in later rounds indicates that model alignment is progressively established across all clients.
> > >
> > >
> > > **Q3:** Parameter sensitivity discussion.
> > >
> > > **A3:** Following your advice, we conduct a multi-factor orthogonal sensitivity study under the suggested topology, using ResNet-18 on CIFAR-10 with $\alpha=0.1$.
> > > The results are shown in the below Table. We observe that the performance variation of DecFus remains limited across different hyperparameter settings, even under this relatively complex topology. This further verifies the effectiveness and robustness of DecFus.
> > >
> > > ||$\epsilon_s$|$\alpha$|$\beta$|$\kappa$|$\gamma$|Accuracy
> > > |-|-|-|-|-|-|-
> > > |1|0.33|15|0.35|0.45|0.35|27.12±0.63
> > > |2|0.33|25|0.45|0.55|0.45|26.83±0.48
> > > |3|0.33|15|0.35|0.55|0.45|25.65±0.34
> > > |4|0.33|25|0.45|0.45|0.35|25.39±0.65
> > > |5|0.37|15|0.45|0.45|0.45|24.74±0.77
> > > |6|0.37|25|0.35|0.55|0.35|24.42±0.45
> > > |7|0.37|15|0.45|0.55|0.35|25.52±0.81
> > > |8|0.37|25|0.35|0.45|0.45|24.89±1.12
> > > |9(Default)|0.35|20|0.40|0.50|0.40|27.86±0.30

---

### Official Review · Reviewer_9kMC · 2026-03-13

**Soundness:** 2
**Presentation:** 3
**Significance:** 2
**Originality:** 2
**Overall Recommendation:** 4
**Confidence:** 4

**Summary:**

This paper addresses the "stuck-at-local-search" problem in decentralized federated learning. The authors propose DecFus, a novel layer-wise fusion framework that unifies layer exchange for exploration and layer averaging for exploitation. The paper introduces a stability-triggered dynamic transition and a similarity-based grouping strategy, and the proposed method effectively navigates complex non-convex loss landscapes.

**Compliance With Llm Reviewing Policy:**

Affirmed.

**Final Justification:**

I believe the authors have addressed all my questions. I will raise my score to 4.

**Key Questions For Authors:**

The majority of the concerns are outlined in the 'Weaknesses' section.

**Limitations:**

I think the authors should include discussions on limitations related to hyper-parameter sensitivity, privacy concerns, and practical use cases, etc.

**Strengths And Weaknesses:**

Strengths:
* The paper is based on a strong and clear technical motivation, as the empirical observations show that vanilla averaging ensures stability but limits exploration.
* The paper establishes clear convergence guarantees without relying on the restrictive "doubly stochastic" aggregation matrix assumption**, which** is very commonly used in existing DFL literature.
* Extensive empirical validation against several SOTA baselines across several settings makes the empirical results solid as well.

Weaknesses:
* FedMR (Hu et al., 2024) has proposed layer-wise model recombination. The “DFLMR” in Table 1 seems to be the DFL version of FedMR, and I think the difference between applying FedMR to decentralized settings and the proposed DecFus is not clear.
* The dynamic cutoff determination mechanism relies on several parameters, such as $\epsilon_s, \alpha, \beta, \kappa$. While these parameters mostly remain fixed across datasets, I think it would be better if there were a more detailed discussion on their sensitivity.
* I am not entirely sure about certain concepts, such as why Equation (7) defines layer similarity $\sigma_{i,r}$ as the minimum value (worst-case) among all neighbor similarities. There is no theoretical or experimental basis for this choice.
* About the theoretical assumptions, I am still not sure whether Assumption 5.4 is feasible in real-world settings.
* Theorem 5.8's convergence bound includes a term of the form $(1-\sqrt{\tilde{\rho}})^{-2}$. As $\tilde{\rho} \to 1$ (slow mixing), this term tends to infinity, yet the range and actual values of $\tilde{\rho}$ are not subject to specific analysis.
* Is Figure 1 a schematic loss surface diagram or an actual loss surface diagram? It is used to claim that “layer exchange leads to instability,” but if it is not an actual experimental result, the argument remains weak.

---

> ### Author Rebuttal · Authors · 2026-03-30
>
> We appreciate your good questions and insightful comments.
>
> **Q1:** Difference between DFLMR and DecFus.
>
> **A1:** From the algorithmic perspective, DFLMR adopts a layer exchange–only strategy, while DecFus proposes a unified layer-wise fusion strategy. Specifically, DecFus dynamically balances layer exchange for exploration and layer averaging for exploitation. Notably, DFLMR can be viewed as a special case of DecFus with the dynamic cutoff fixed at 1, reducing the layer-wise fusion to pure layer exchange.
>
> From the theoretical perspective, we establish convergence guarantees for a more general aggregation process underlying DecFus, without relying on the common doubly stochastic assumption of the aggregation matrix in existing DFL convergence analysis.
>
> In summary, DecFus is a distinct DFL framework in both algorithm and theory, rather than a simple extension of FedMR.
>
> **Q2:** Parameter sensitivity discussion.
>
> **A2:** To further verify the sensitivity of dynamic cutoff parameters, we conducted a multi-factor orthogonal sensitivity study in setting (SVHN, VGG16, $\alpha=1$) as follows.
> ||$\epsilon_s$|$\alpha$|$\beta$|$\kappa$|$\gamma$|Accuracy
> |-|-|-|-|-|-|-
> |1|0.33|15|0.35|0.45|0.35|93.44±0.30
> |2|0.33|25|0.45|0.55|0.45|92.98±1.36
> |3|0.33|15|0.35|0.55|0.45|93.22±1.12
> |4|0.33|25|0.45|0.45|0.35|93.73±0.65
> |5|0.37|15|0.45|0.45|0.45|91.84±1.80
> |6|0.37|25|0.35|0.55|0.35|93.83±1.81
> |7|0.37|15|0.45|0.55|0.35|93.30±0.57
> |8|0.37|25|0.35|0.45|0.45|92.39±2.56
> |9|0.35|15|0.35|0.50|0.40|93.23±0.88
> |10|0.35|25|0.45|0.50|0.40|92.49±3.29
>
> It show that varying these hyperparameters leads to less than $2\%$ accuracy fluctuation, while DecFus consistently outperforms most baselines.
>
> These parameters mainly govern the transition from exploration-dominant exchange to exploitation-dominant averaging, and do not significantly affect the core mechanism. As a result, DecFus does not rely on careful hyperparameter tuning. We will include the above sensitivity analysis in the appendix.
>
>
> **Q3:** The choice of $\sigma_{i,r}$ in Equation (7).
>
> **A3:** The $\sigma_{i,r}$ highlights the most mismatched neighbor, capturing the diversity DecFus preserves for exploration. In contrast, mean or maximum similarity can be dominated by well-aligned neighbors, masking informative disagreement. This aligns with the exchange mechanism, which prioritizes locally hard-to-align layers. Furthermore, we added additional ablation experiments under the setting (SVHN, ResNet-50), where the minimum in Eq.~(7) is replaced by the mean and maximum similarity.
> ||Min (ours)|Max|Mean
> |--|--|--|--
> |IID|93.43±0.04|92.24±0.11|92.38±0.05
> |non-IID $\alpha=0.1$|57.83±0.85|53.75±1.57|54.19±1.55
>
> The results show that the minimum definition achieves the best overall performance, supporting our claims.
>
> **Q4:** The feasibility of Assumption 5.4.
>
> **A4:** Assumption 5.4 formalizes a mild and practical requirement, where each layer participates in averaging in at least one training round. This aligns with DecFus, where the dynamic cutoff gradually shifts training from exploration-dominant exchange to exploitation-dominant averaging, reducing the proportion of exchanged layers over time. As a result, persistent exchange becomes unlikely in later stages, making Assumption 5.4 consistent with the algorithm’s intended behavior rather than an unrealistic constraint.
>
> **Q5:** Specific analysis on $\tilde{\rho}$.
>
> **A5:** From the equation $\widetilde{\rho^t}=\max\limits_r(\rho_r^t)\left(1-q(1-N\epsilon)^2\right)^{m}\in[0,1)$ below Eq.~(13), we observe that the magnitude of $\widetilde{\rho^t}$ is governed by two factors: $\rho_r^t$ and $\left(1-q(1-N\epsilon)^2\right)^m$. In particular, $\widetilde{\rho^t} \rightarrow1$ can only occur when $\left(1-q(1-N\epsilon)^2\right)^m = 1$ and $\rho_r^t\rightarrow1$. However, for a doubly stochastic matrix that is irreducible and aperiodic, it strictly holds that $\rho_r^t<1$ and does not approach 1. This is a direct consequence of the spectral gap property of such matrices. Moreover, a similar denominator term appears in [1], where this spectral gap property is rigorously characterized.
>
>
> **Q6:** Clarification on Figure 1.
>
> **A6:** Figure 1 is a conceptual illustration, which is to provide readers an intuitive depiction of the three aggregation behaviors prior to presenting empirical evidence. The empirical support is provided in Figure 2, which show that layer averaging consistently yields smaller shifts, while layer exchange produces significantly larger shifts that remain non-negligible even in later stages. In contrast, DecFus exhibits intermediate behavior: larger shifts in early rounds and smaller shifts in later rounds, aligning with the intended transition from exploration to exploitation.
>
> [1] X. Lian, C. Zhang, H. Zhang, C.-J. Hsieh, W. Zhang, and
> J. Liu, Can Decentralized Algorithms Outperform Centralized Algorithms? A Case Study for Decentralized Parallel Stochastic Gradient Descent, NeurIPS, 2017.

---

> > ### Author Rebuttal · Reviewer_9kMC · 2026-04-03
> >
> > Thank the authors for the detailed and well-structured response. I believe the authors have addressed all my questions. I would encourage them to explicitly incorporate these clarifications in the revised version. I will raise my score to 4.

---

> > > ### Author Response · Authors · 2026-04-07
> > >
> > > Thank you for the positive feedback and for your thoughtful comments throughout the review process. We are glad that our response has addressed your questions and helped clarify the contribution of the paper.

---

### Decision · Program_Chairs · 2026-04-30

**Decision:**

Accept (regular)

**Comment:**

This paper addresses an important limitation of decentralized federated learning, namely the performance ceiling induced by pure model averaging under non-IID data and sparse communication, and proposes DecFus, a unified layer-wise fusion framework that combines layer exchange for exploration with layer averaging for exploitation. The authors strive to present the central question of how to balance these two behaviors both across training stages and across layers, and the resulting method is technically meaningful. Although reviewers raised reasonable questions about sensitivity, and additional overhead, the rebuttal substantially clarified the distinction from prior layer-exchange methods, added sensitivity evidence, justified the layer-similarity design, and provided broader evidence of generality beyond vision tasks, which resolved the main concerns for positive reviewers. Overall, I therefore recommend accept.